# Molecular insights into substrate translocation in an elevator-type metal transporter

Yao Zhang [1,7], Majid Jafari[1,7], Tuo Zhang [1,2,7], Dexin Sui[1], Luca Sagresti [3,4], Kenneth M. Merz Jr. [1,5,6] ✉ & Jian Hu [1,5] ✉

The Zrt/Irt-like protein (ZIP) metal transporters are key players in maintaining the homeostasis of a panel of essential microelements. The prototypical ZIP from *Bordetella bronchiseptica* (BbZIP) is an elevator transporter, but how the metal substrate moves along the transport pathway and how the transporter changes conformation to allow alternating access remain to be elucidated. Here, we combine structural, biochemical, and computational approaches to investigate the process of metal substrate translocation along with the global structural rearrangement. Our study reveals an upward hinge motion of the transport domain in a high-resolution crystal structure of a cross-linked variant, elucidates the mechanisms of metal release from the transport site into the cytoplasm and activity regulation by a cytoplasmic metal-binding loop, and unravels an unusual elevator mode in enhanced sampling simulations that distinguishes BbZIP from other elevator transporters. This work provides important insights into the metal transport mechanism of the ZIP family.

A panel of first-row *d*-block metal elements are essential for life. Among the transporters that control the fluxes of these micronutrients across biological membranes, the Zrt/Irt-like protein (ZIP) family plays a central role[1–3]. The ZIP family consists of divalent metal transporters that are ubiquitously expressed in all kingdoms of life, mediating metal flux into the cytoplasm from the extracellular milieu or from the intracellular organelles/vesicles. In humans, the fourteen ZIPs (ZIP1-14) are involved in zinc, iron, or manganese homeostasis. Some ZIPs are associated with human diseases[4–9] and are therefore considered as potential targets for therapies[10–18]. For instance, ZIP6 is tightly associated with cancer cell growth and proliferation[12,19–21], and a monoclonal antibody against human ZIP6 is being tested in clinical trials for breast cancer and other solid tumors[22,23]. ZIP4, which was found to be aberrantly upregulated in many cancers, has been shown to be associated with cancer cell growth and metastasis[24–32], whereas the ER-residing ZIP7 is essential for B cell development[33] and a druggable node in the Notch pathway[17].

Recent structural studies have begun to elucidate the transport mechanism of ZIPs. The structure of a prototypic ZIP from *Bordetella bronchiseptica* (BbZIP) revealed the structural framework of the conserved transmembrane domain that is composed of eight transmembrane helices (TM) and two tethered metal binding sites (M1 and M2) at the transport site[34]. More BbZIP structures in the inward-facing conformation (IFC) were later solved by x-ray crystallography and cryo-electron microscopy (cryo-EM)[34–38]. An elevator transport mode has been proposed based on the two-domain architecture, evidence from evolutionary covariance, and an experimentally validated outward-facing conformation (OFC) model generated by repeat-swap homology modeling[36,38]. According to the proposed mechanism, a TM bundle composed of TM1/4/5/6 (the transport domain), carrying the metal substrate(s), moves vertically relative to the static scaffold domain composed of TM2/3/7/8 to achieve alternating access. The structure models predicted by AlphaFold2 suggested that the elevator mode is commonly used by the ZIPs from different subfamilies[38,39]. Due to the

¹Department of Biochemistry & Molecular Biology, Michigan State University, East Lansing, MI, USA. ²College of Food Science and Nutritional Engineering, China Agricultural University, Beijing, China. ³Scuola Normale Superiore, Pisa, Italy. ⁴Istituto Nazionale di Fisica Nucleare (INFN) sezione di Pisa, Pisa, Italy. ⁵Department of Chemistry, Michigan State University, East Lansing, MI, USA. ⁶Center for Computational Life Sciences, Lerner Research Institute, The Cleveland Clinic, Cleveland, OH, USA. ⁷These authors contributed equally: Yao Zhang, Majid Jafari, Tuo Zhang. ✉e-mail: merz@chemistry.msu.edu; hujian1@msu.edu

lack of structural similarity to other well-characterized elevator transporters[40], a recent structural survey did not consider BbZIP to be a classical elevator transporter, suggesting that the mechanism may be fundamentally different from what has been established for elevator transporters. Therefore, it is imperative to elucidate the detailed processes of conformational change and translocation of metal substrate through the transporter.

In this work, we integrated structural, biochemical, and computational approaches to address these critical questions. A high-resolution crystal structure of an Hg²⁺-crosslinked BbZIP variant revealed an upward hinge motion of the transport domain and identified a third metal binding site at the end of the transport pathway. Structural analysis and functional studies allowed us to propose mechanisms for metal release from the transport site into the cytoplasm and for activity regulation via a cytoplasmic loop containing the third metal binding site. Our enhanced sampling simulations not only supported these proposed mechanisms but also captured the process of the elevator motion. The combined experimental and computational data revealed an unprecedented mode of transport for elevator transporters.

## Results

### An upward hinge motion of the transport domain

Our previous study has suggested that when the transport domain moves upward against the scaffold domain to switch from the IFC to the OFC, the distances between some residue pairs in the IFC become shorter in the OFC[38], such as A95 and A214 located on the scaffold and transport domains, respectively (Fig. 1A). This represents the dynamics of the transporter, which is essential for the alternating access mechanism, but it also poses a challenge for structural studies. To fix the relative orientation of the two domains for structural characterization, A95 and A214 were substituted by cysteine residues, allowing chemical cross-linking between them as there is no endogenous cysteine residue in BbZIP. To promote disulfide bond formation, the membrane fraction of the *Escherichia coli* cells expressing the A95C/A214C variant was treated with [Cu(II)(1,10-phenanthroline)₃] at room temperature and then applied to Western blot to detect the cross-linked species. As shown in Fig. 1B, the treatment led to the generation of a band with reduced migration in the non-reducing SDS-PAGE, which was eliminated upon the addition of a reducing agent 2-mercaptoethanol. The same treatment of wild-type BbZIP, A95C, or A214C single variants did not result in band shift, indicating that a disulfide bond has been formed between C95 and C214. Given that the Cα distance between A95 and A214 (10.1 Å) in the IFC, as shown in the cadmium (Cd) bound structure (PDB: 5TSB, Fig. 1A), is longer than that between two disulfide bonded cysteine residues (3.0–7.5 Å[41]), the formation of the C95-C214 disulfide bond is consistent with the proposed elevator transport mode[36,38]. The inability to completely convert the A95C/A214C variant to the cross-linked form, presumably due to the highly dynamic nature of the transporter, led us to use mercury (Hg) as a cross-linker for structural study. As shown in Fig. 1C, addition of HgCl₂ to the purified A95C/A214C variant caused a clear band shift in the non-reducing SDS-PAGE only for the double variant but not for any of the single variants (C95 and C214). The band shift was prevented by the pretreatment of the purified protein with N-ethylmaleimide (NEM), a thiol reacting agent, and completely abolished by 2-mercaptoethanol, supporting that the two introduced cysteine residues have been cross-linked by Hg²⁺.

After a brief treatment with EDTA and desalting, the Hg²⁺-crosslinked sample was crystallized in lipidic cubic phase under a Cd-free

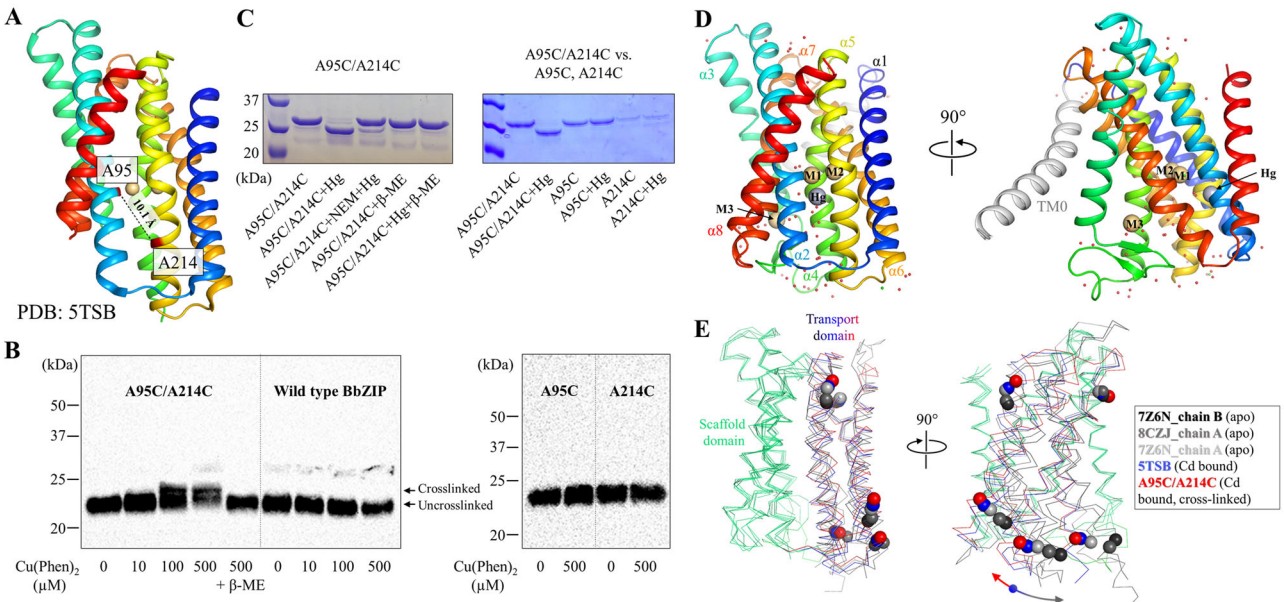

**Fig. 1 | Chemical cross-linking and structure determination of the A95C/A214C variant. A** A95 and A214 in IFC of BbZIP (PDB: 5TSB). The distance between the Cα atoms (10.1 Å) is greater than the distance required for a disulfide bond (3.0–7.5 Å, ref.[41]). **B** Disulfide bond cross-linking of the variants of A95C/A214C, A95C, A214C, and wild-type BbZIP in the membrane fraction of the *E.coli* cells overexpressing the target proteins by [Cu(II)(1,10-phenanthroline)₃] at the indicated concentrations. BbZIP and the variants were detected in Western blot by using a custom mouse monoclonal antibody against BbZIP. At least three independent experiments were conducted with similar results. **C** Cross-linking of the purified variants of A95C/A214C, A95C, and A214C by HgCl₂ at the protein:Hg molar ratio of 1:10. Pretreatment with N-ethylmaleimide (NEM) and post-treatment with 2-mercaptoethanol (β-ME) prevented and reversed the cross-linking reaction, respectively. At least three independent experiments were conducted with similar results. **D** Overall structure of the Hg²⁺-cross-linked A95C/A214C variant. The conserved eight TMs are colored in rainbow and TM0 is in grey. Cd²⁺ at the M1, M2, and M3 sites are depicted as light-brown spheres, and Hg²⁺ is shown as a grey sphere. **E** Hinge motion of the transport domain. The scaffold domains of BbZIP (green) in representative structures are aligned with selected Cα atoms shown in sphere mode. When compared to the Cd-bound state (PDB: 5TSB, blue), the transport domain of the cross-linked variant (red) undergoes an upward/clockwise hinge motion, whereas those in the apo state structures (grey to black) rotate in the opposite direction. The arrows indicate the opposite directions of the hinge motion from the Cd-bound state only to highlight the new conformation of the cross-linked structure when compared to the previously solved structures.

condition, yielding a structure of BbZIP in a distinct IFC conformation at 1.95 Å, the highest resolution for BbZIP to date (Fig. 1D and Supplementary Table 1). Similar to the structures of BbZIP in the apo state[36,38], the structure showed nine TMs with the N-terminal TM0 only weakly associated with TM3 and TM6. As expected, a $Hg^{2+}$ is chelated by C95 from TM2 and C214 from TM5 (Fig. 1D and Supplementary Fig. 1), locking the relative orientation between the transport and scaffold domains. When compared with the previously solved structure of BbZIP in the Cd-bound state (PDB: 5TSB), the structure revealed an upward rotation of the transport domain relative to the scaffold domain (Fig. 1E and Supplementary Fig. 2), which is contrast to the structures in the metal-free state that consistently showed a rotation of the transport domain in the opposite direction[36,38]. Of great interest, when the scaffold domains of the representative IFC structures, including the structure, are aligned, a continuous rigid-body hinge motion of the transport domain can be visualized (Fig. 1E), which was recapitulated in the computational simulations shown in later sections.

## Characterization of the high-affinity metal binding sites

Although the cross-linked A95C/A214C variant was briefly treated with EDTA and crystallized in a $Cd^{2+}$-free buffer, the two tethered metal binding sites (M1 and M2) are still occupied by $Cd^{2+}$ (Fig. 2A). The coordination spheres of the bound $Cd^{2+}$ are not significantly affected by the rigid-body hinge motion of the transport domain, except that a density 2.6 Å away from the $Cd^{2+}$ at the M1 site was found (Supplementary Fig. 3). This density, which excludes the sulfur atom of M99 from the coordination sphere of the M1 metal (Fig. 2A), was assigned to be a chloride ion ($Cl^-$) based on the crystallization condition and the surrounding contacts, but we cannot rule out the possibility that it may be a different $Cd^{2+}$ ligand. In the structure, a third metal binding site (M3) was identified at the end of the metal release tunnel. The $Cd^{2+}$ bound at the M3 site, with an occupancy of 0.84, is coordinated with two histidine residues (H149 and H151) from the intracellular loop 2 (hereafter IL2), D144 from TM3, E276 from TM7, and an ordered water molecule, forming a distorted octahedral coordination sphere (Fig. 2B and Supplementary Fig. 4). Recently, a cryo-EM structure of wild-type BbZIP showed a similar M3 binding site[37], and structural comparison of the two structures revealed some minor differences (Supplementary Fig. 2C), which is consistent with a flexible IL2 and dynamic metal binding at the M3 site. To examine whether M1−M3 are an authentic metal binding sites in solution, wild-type BbZIP and the H149A/H151A variant with the eliminated M3 site purified in the presence of 0.25 mM $Cd^{2+}$ were applied to a size-exclusion chromatography column equilibrated with a Cd-free solution, and the peak fractions were analyzed by using inductively coupled plasma mass spectrometry (ICP-MS) to quantify the bound metal. The determined Cd/BbZIP molar ratio was $3.0 \pm 0.3$ and $1.8 \pm 0.3$ (mean ± s.e.m., n = 2, Fig. 2C) for wild-type protein and the H149A/H151A variant, respectively, indicating that $Cd^{2+}$ is indeed bound at the M3 site in solution. Although the IL2 of BbZIP was found to be severely disordered in the previous crystal structures, $Cd^{2+}$ binding to the M3 site stabilizes the IL2 which folds into two short antiparallel β-strands and forms hydrogen bonds with the rest of the protein. The large B-factor of the $Cd^{2+}$-stabilized IL2 is consistent with the notion that it is intrinsically disordered (Supplementary Fig. 5). As a result, the metal release tunnel in the structure, starting from the M1 site, is divided into two parallel pathways – one is connected to the M3 site (Path 1), whereas the other, filled with water molecules, is directly connected to the cytoplasm (Path 2) (Fig. 2D).

## A conserved metal release mechanism

To understand the function of the M3 site, we studied the residues in human ZIP4 (hZIP4) that are topologically equivalent to those forming the M3 site in BbZIP (Fig. 2E). It is known that most ZIPs have a histidine-rich segment in the IL2[2,3], and D144 and E276 are generally conserved in ZIPs from different subfamilies and different species

(Supplementary Fig. 6). E417 in hZIP4, which is equivalent to D144 in BbZIP, was found to be dispensable for activity (Fig. 2F) as the E417A variant exhibited a modestly increased $V_{max}$ when compared to wild-type hZIP4. Different from the short IL2 of BbZIP, the corresponding loop of hZIP4 is longer and harbors a His-rich cluster containing five histidine residues. We replaced all the histidine residues with alanine to generate the 5HA variant which exhibited a drastically changed transport kinetics (Fig. 2G). The dose-dependent profile appeared to be linear at low zinc concentrations and slightly bent only at high zinc concentrations. Because the curve did not saturate in the range of zinc concentrations tested in this work, we could not fit the curve to obtain the values for $K_M$ and $V_{max}$, but the curve profile suggested that both $K_M$ and $V_{max}$ increased. The results of the variants of E417A and 5HA suggested that the residues corresponding to those forming the M3 site in BbZIP appear to negatively regulate the activity of hZIP4. Consistently, replacing H149 and H151 with alanine led to an increased activity of BbZIP[37]. Unexpectedly, D604 in hZIP4, which is equivalent to E276 in the M3 site, is indispensable for the transport activity. Substitution of D604 with alanine completely abolished the zinc transport activity (Fig. 2H), and only the D604E variant, but not the D604H variant, preserved an activity similar to that of wild-type hZIP4 (Fig. 2I). Notably, E276 in human ZIP2, the D604-equivalent residue, was also shown to be absolutely essential for the transport activity[42].

To understand the role of E276 in BbZIP and the equivalent residues in other ZIPs, multiple metal bound structures of BbZIP (PDB: 5TSA and 7Z6M, and the $Hg^{2+}$-cross-linked structure) were superimposed, revealing potential metal release pathways from the transport site to the cytoplasm where E276 plays a role in the metal relay (Fig. 2K). Since the M2 site has no direct access to the cytoplasm (blocked by I174 and V215, Fig. 2D)[35], metal release from the transport site likely starts from the M1 site. Swinging of the side chain of H177 away from the M1 site opens a door for metal to leave the high-affinity transport site. Joining the flipped H177, E276 forms a transient metal binding site and then undergoes a nearly right-angle rotation to direct the metal substrate to the M3 site (Path 1) where the metal is retained before being released into the cytoplasm. Therefore, E276 appears to play multiple roles in the process of metal release from the transport site to the cytoplasm. First, it transiently holds the metal that leaves the M1 site with the flipped H177. This role has been suggested by our and other previous studies[34,36,37]. Second, its structural flexibility, likely due to P279, which is present in many ZIPs (Supplementary Fig. 6), and the high dynamics of the segment connecting TM7 and TM8 (Supplementary Fig. 5), allows it to direct the metal to the M3 site. Indeed, replacing P607 in hZIP4 (equivalent to P279 in BbZIP) with alanine reduced the transport activity by ~60% (Fig. 2J). Third, it contributes to metal binding to the M3 site. As a high-affinity metal binding site (Fig. 2C) that is within the transport pathway and negatively regulates the metal transport activity (Fig. 2F, G), the M3 site can be described as a metal sink in addition to the proposed autoinhibitory site that acts only upon zinc overload[37]. As Path 2 is filled with water (Fig. 2D), can it be used as an alternative pathway for metal release to the cytoplasm when Path 1 is blocked by the occupied M3 site? We investigated this important question using computational approaches.

## Metal release and elevator motion revealed in simulations

Next, we deployed enhanced sampling simulations, i.e., metadynamics (MTD) simulations in this work, to study the process of metal release from the transport site into the cytoplasm. MTD simulations have been used to study substrate binding and release in various transporters[43–46], but have not been applied to any highly charged system like BbZIP due to the challenges in simulating the behaviors of transition metals. Established on our recent work on zinc binding to various ligands[47,48], we systematically identified the optimal configuration of parameters governing the system dynamics and addressed the pressing issue of zinc translocation through BbZIP by using

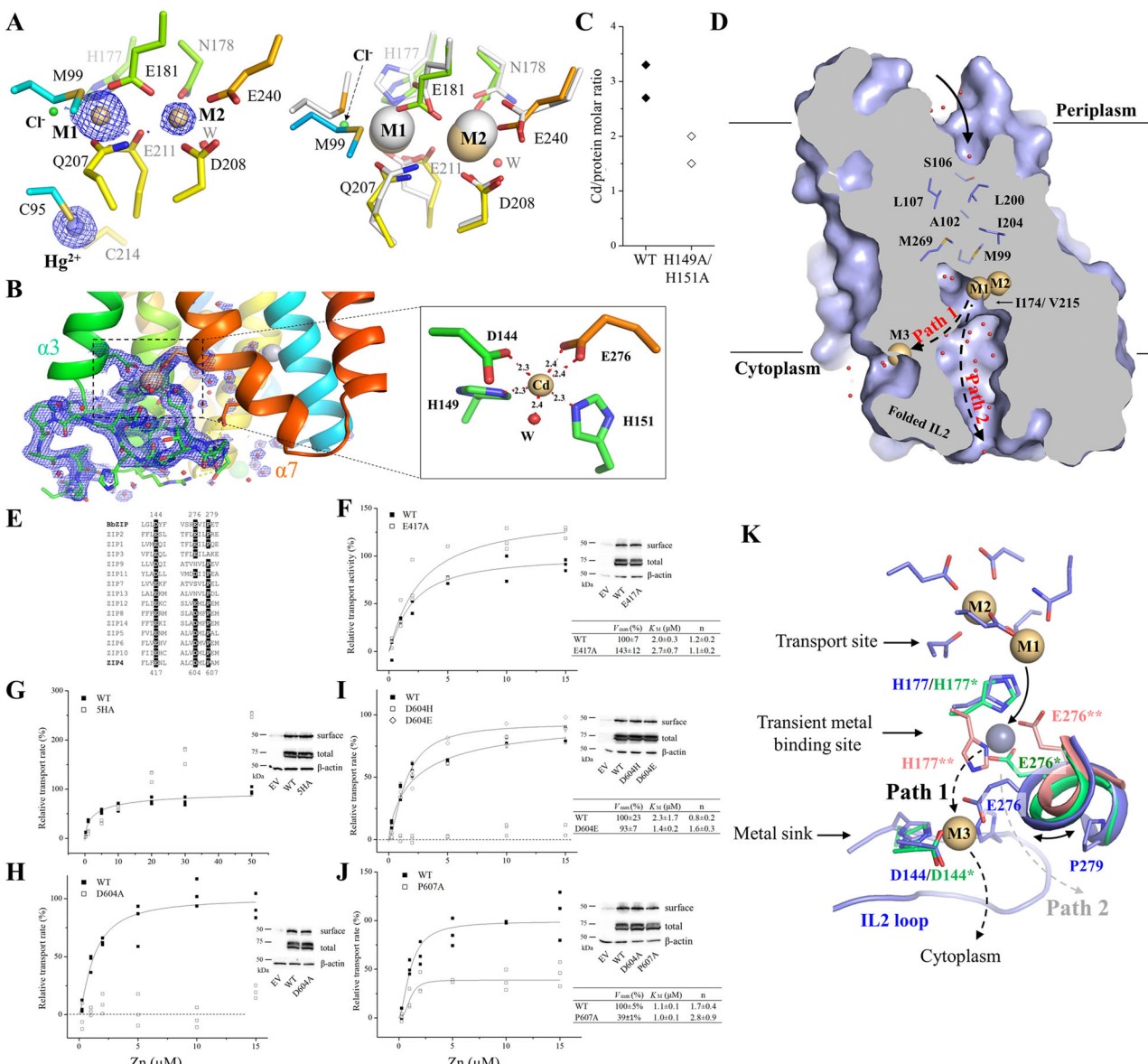

**Fig. 2 | Metal binding to BbZIP and metal release from the transport site.**
**A** Metal binding at the transport site of the A95C/A214C variant. *Left*: $Cd^{2+}$ binding at the M1 and M2 sites and $Hg^{2+}$ binding between C95 and C214. The blue meshes indicate the electron densities of $Cd^{2+}$ and $Hg^{2+}$ (2Fo-Fc map, σ = 4). The occupancies are 0.85 and 0.42 for $Cd^{2+}$ at M1 and M2, and 0.36 for $Hg^{2+}$, respectively. *Right*: structural comparison of the transport site in the structure of the variant (rainbow color) with that of the $Cd^{2+}$-bound structure (white, PDB: 5TSB). **B** The structure of the IL2 and the M3 site. The blue meshes indicate the 2Fo-Fc map (σ = 1) of the IL2 and the ordered water molecules (red balls). The occupancy of $Cd^{2+}$ at the M3 site is 0.84. The hydrogen bonds involved in stabilizing the IL2 are shown as yellow dashed lines. The inset shows the zoom-in view of $Cd^{2+}$ binding at the M3 site with indicated distances between $Cd^{2+}$ and the coordinating atoms (in angstrom). **C** $Cd^{2+}$/protein molar ratios of wild-type BbZIP (solid rhomboids) and the H149A/

H151A variant (open rhomboids) from two independent experiments. **D** Divided metal release tunnel by the folded IL2. **E** Sequence alignment of BbZIP and human ZIPs to highlight D144, E276, and P279. **F**–**J** Kinetic study of the hZIP4 variants in the cell-based $^{65}Zn$ transport assay. The dose curves were fitted with the Hill model, and the kinetic parameters with standard errors obtained from curve fitting are listed in the tables. The data shown are the results from one of three independent experiments and three biological replicates (n = 3) were included for each condition in one experiment. The total and cell surface expression of hZIP4 and its variants were detected in Western blot using an anti-HA antibody. **K** Metal release pathways. The new structure (blue), the Cd-bound structure (PDB: 7Z6M, green), and the zinc-substituted structure (PDB: 5TSA, pink) are superimposed to show Path 1 (primary, black dashed arrows) and Path 2 (alternative, grey dashed arrow) for metal release from the transport site into the cytoplasm.

volumetric collective variables (CVs) and iteratively optimizing the MTD simulation parameters.

The cryo-EM structure of wild-type BbZIP (PDB: 8GHT) was used as the initial structural model in the MTD simulations because it is the only experimentally solved structure of a BbZIP dimer[37], of which the dimerization interface has been validated by chemical crosslinking[38]. The cryo-EM structure of the BbZIP is highly superimposable on the crystal structure of BbZIP in the Cd-bound state (PDB: 5TSB) (Supplementary Fig. 7A). To generate the initial structural model for

simulations, $Cd^{2+}$ ions at the M1-M3 sites in the cryo-EM structure were replaced by $Zn^{2+}$, which did not lead to significant structural changes after reaching equilibrium (Supplementary Fig. 7B). When standard molecular dynamics (MD) simulations were performed on the Zn-bound structural model for up to 2 μs, no significant structural change or metal movement was noticed, indicating that the Zn-bound state represents a stable conformation and also necessitating the use of MTD simulations to accelerate the process through enhanced sampling. In MTD simulations, a biasing potential was applied to the zinc

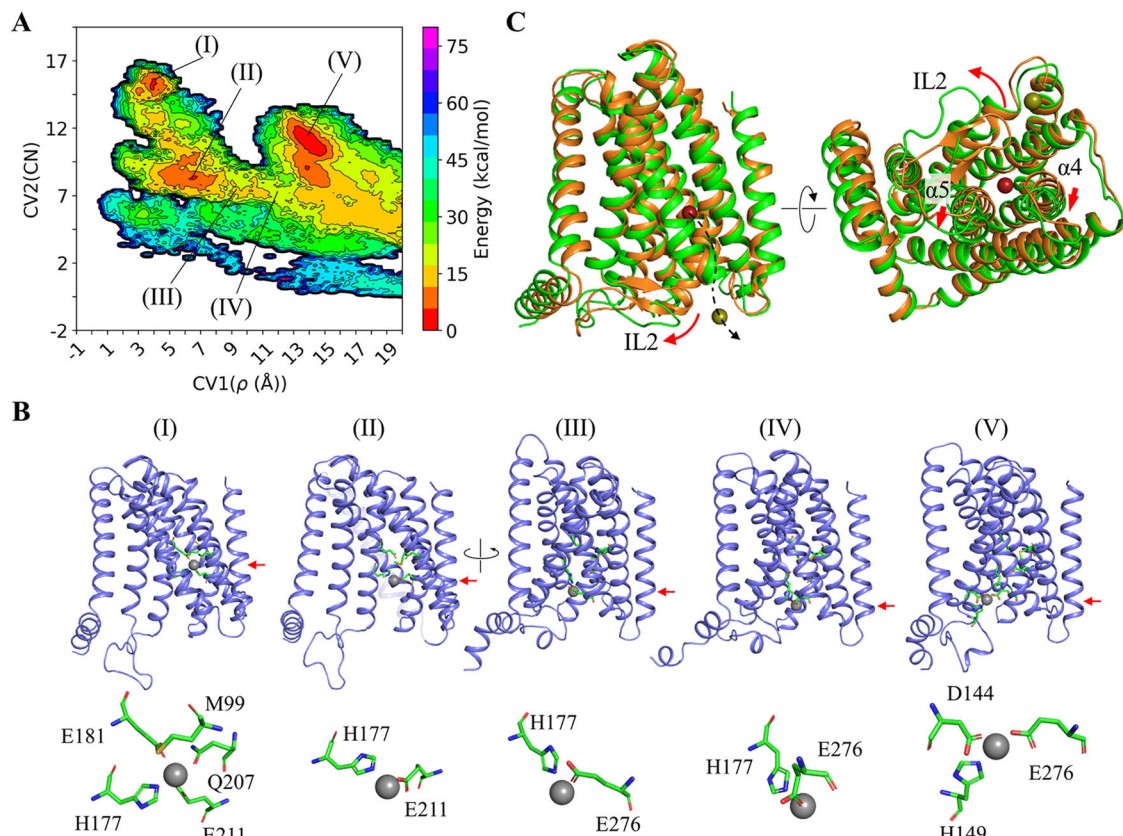

**Fig. 3 | Metal release from the M1 site into the cytoplasm revealed by MTD simulations. A** Free energy surface map of metal coordination upon metal release via Path 1 in the first scenario. CV1 represents the distance from the metal ion to the protein center of mass, and CV2 is the number of non-hydrogen atoms within 4 Å of the metal. The shown data are from one out of three independent simulations. The results of the other two simulations are shown in Fig. S8A. **B** Snapshots of the overall structure of the transporter and metal coordination at the selected local minima in (**A**). The red arrows indicate the vertical positions of the metal when it

moved from the M1 site to the M3 site via Path 1. In State V, the metal is bound at the M3 site. **C** Structural rearrangements (red arrows) of the IL2 and the cytoplasmic portions of TM4 and TM5 upon metal released into the cytoplasm via Path 2 (dashed curve arrow) in the second scenario. The structural modes before and after metal release into the cytoplasm are colored in orange and green, respectively. The shown data are from one out of four simulations. The results of the other three simulations are shown in Fig. S8B.

ion bound at the M1 site because the M1 site has been shown to be the authentic transport site[35].

Given that the process of metal release may be affected by the folding state of the IL2 (Fig. 2D) and the metal binding states of the transporter, we conducted simulations in the following four scenarios. In the first scenario, the M1 and M2 sites were occupied by $Zn^{2+}$ but the IL2 was unfolded with no $Zn^{2+}$ bound at the M3 site, mimicking the conformational states shown in the Cd-bound crystal structures (PDB: 5TSB and 7Z6M). To unfold the IL2, a steered MD simulation was conducted to generate the initial model. In the second scenario, the M1-M3 sites were occupied by $Zn^{2+}$ and the IL2 was in the folded state, mimicking the conformational state solved in the cryo-EM structure (PDB: 8GHT) and the structure reported in this work. In the third scenario, the M1 and M2 sites were occupied by $Zn^{2+}$ and the IL2 was in the folded state with a pre-formed but empty M3 site to test the accessibility of metal to an unoccupied M3 site upon a folded IL2. In the last scenario, the settings are the same as in the second scenario except that the metal at the M2 site was removed to study the role of the metal bound at the M2 site in metal release. The settings and results of all MTD simulations are summarized in Supplementary Table 2.

In the first scenario, three independent runs of MTD simulations revealed multiple neighboring local minima in the free energy surface maps (Fig. 3A and Supplementary Fig. S8A), which allowed us to construct a pathway for metal release from the M1 site to the cytoplasm (Fig. 3B). It starts with the flip of the H177 side chain away from the

M1 site, as observed in our early computational work[49], and the metal is then coordinated by both H177 and E276 at the transient $Zn^{2+}$ binding site, followed by its movement to the M3 site. Of great interest, this process is essentially in agreement with the proposed Path 1 based on the experimentally solved structures (Fig. 2K). A notable difference is that $Zn^{2+}$ at the M3 site was coordinated with only three residues (D144 plus two of H149, H177, H275, and E276) while H151 in the IL2 did not join the M3 site in any simulation. Although it is possible that the folding of the 18-residue IL2 takes a longer time than the simulation length to allow H151 to join the M3 site, another possibility that H151 really does not participate in metal binding during the metal release process cannot be excluded. Nevertheless, our MTD simulations indicated that (1) E276 is directly involved in the processes of metal release from the M1 site and metal delivery to the M3 site; and (2) the metal from the transport site can be retained at the M3 site in its journey to the cytoplasm as the M3 site is a significant local minimum in the free energy surface map.

In the second scenario, four out of five simulations showed that $Zn^{2+}$ remained at the M1 site until the biasing potential pushed it out of the high-affinity binding site before it was finally released into the cytoplasm (Supplementary Table 2). Since the IL2 was in the folded state and the M3 site was occupied throughout the simulations, the metal from the M1 site passed through Path 2 to reach the cytoplasm (Fig. 2D, K). However, this occurred only when the conformational changes of the portions of TM4 and TM5 on the cytoplasmic side and the IL2 took place to create a larger space for the metal to pass through

(Fig. 3C and Supplementary Fig. 8B). Considering the requirement for a significant structural rearrangement of multiple structural elements, metal passing through Path 2 is likely to be kinetically unfavorable compared to Path 1. This result is consistent with the suggested role of the M3 site as an autoinhibitory zinc sensor that abolishes the zinc transport activity when the M3 site is occupied upon intracellular zinc overload[37].

Unexpectedly, for one simulation in the second scenario, $Zn^{2+}$ at the M1 site was found to move backward and eventually released into the periplasm. The RMSF analysis showed that the residues in the transport domain exhibit greater structural fluctuation than those observed in the first scenario (Supplementary Fig. 9), whereas the intracellular loops, including the IL2, showed smaller structural fluctuation due to the folded and stabilized IL2. Close inspection of the conformational change revealed that TM4 and TM5, and especially the latter, underwent a significant upward movement relative to the scaffold domain, along with a swing of the portion of TM5 on the periplasmic side away from the tunnel (Fig. 4A, B), opening a path for the metal to be released from the M1 site into the periplasm. Concomitantly, the portion of TM5 on the cytoplasmic side moved toward the tunnel to narrow the pathway toward the cytoplasm (Fig. 4A). This large structural rearrangement of TM4 and TM5 and the rotation of TM1 and TM6 (Fig. 4C) resulted in an OFC-like conformation with the transport site lifted by ~6 Å toward the periplasm, which is a hallmark of elevator transporters, allowing the metal to be released into the periplasm when this major conformational change took place. Analysis of the changes in distance between the residues within and between the transport and scaffold domains provided further support for the proposed elevator motion. For instance, the Cα distance between A95 (TM2) on the scaffold domain and A214 (TM5) on the transport domain changed from ~9 Å to ~5 Å when the major conformational change occurred (at ~180 ns as indicated in Fig. 4D), which is short enough to allow the disulfide bond formation for the A95C/A214C variant (Fig. 1B). Similarly, the distance between S106 (TM2) on the scaffold domain and I196 (TM5) on the transport domain also increased at ~180 ns, indicative of the vertical movement and the outward swing of the periplasmic side of TM5 during the conformational transition (Fig. 4D). In contrast, the distances between the residues within the transport domain (A71 on TM1 and V234 on TM6) or the scaffold domain (S106 on TM2 and M301 on TM8) varied within a narrow range during the simulations (Fig. 4D). The large vertical displacement between the transport and scaffold domains and the much smaller structural fluctuations within each domain indicate a rigid-body motion of the transport domain, which is another hallmark of elevator transporters. Of great interest, the overall conformational change leading to the OFC-like state is essentially a combination of an extrapolation of the hinge motion revealed by the experimentally solved IFC structures (Fig. 1E) and a vertical sliding for the transport domain (Fig. 4E). To the best of our knowledge, such a conformational change has not been observed in other elevator transporters[40,50,51].

Three runs of MTD simulations in the third scenario showed that the M3 site remained empty while the IL2 was in the folded state, indicating that the metal from the M1 site has no direct access to the pre-formed but unoccupied M3 site. This result suggests that, in order to form the state with an occupied M3 site and a fully folded IL2, as observed in the experimentally solved structures, metal binding to the M3 site should precede the folding of the IL2. As Path 1 is blocked under this condition, $Zn^{2+}$ at the M1 site was found to be either released into the cytoplasm via Path 2 or remained near the M1 site (Supplementary Table 2). In one simulation, however, the metal was released into the periplasm when the transport domain underwent a combined hinge motion and vertical sliding, as observed in the second scenario (Supplementary Fig. 10A).

In the last scenario, we tested the role of the M2 site in the process of metal release from the M1 site. The M2 site is present in most ZIPs

but disruption of the M2 site only modestly reduced the zinc transport activity[35]. Only one of five runs of MTD simulations with an unoccupied M2 site showed that the metal from the M1 site was released into the cytoplasm (Supplementary Table 2), in sharp contrast to the second scenario where the metal reached the cytoplasm in four out of five simulations. This result may suggest a role for the M2 metal ($Zn^{2+}$ or a different type of metal[52]) in facilitating metal movement out of the M1 site, which is conceivable given the likely strong electrostatic repulsion between two divalent cations due to the short distance between them (4.4 Å in 5TSB). The metal was released into the periplasm in two simulations where the elevator motion of the transport domain was observed again in a manner similar to what has been observed in the second and third scenarios (Supplementary Fig. 10B, C). Close inspection of the resulting OFC-like structures showed that the cytoplasmic gate of the tunnel was not fully closed. To examine whether the folded IL2 disrupts the closing of the tunnel, we unfolded the IL2 and deleted the metal at the M3 site from the OFC-like model. After running a standard MD simulation for 1 μs, it was found that the movement of the portions of TM4 and TM5 on the cytoplasmic side led to the closing of the cytoplasmic gate composed of a few hydrophobic residues (Supplementary Fig. 11).

Overall, the MTD simulations: (1) established the details of metal release from the transport site into the cytoplasm and the roles of conserved residues (including E276) along the metal release pathway in this process; (2) supported the proposed function of the IL2 as an activity regulator; (3) provided evidence supporting the role of the M2 metal in facilitating metal release from the M1 site; and importantly, (4) revealed an IFC-to-OFC transition through a combination of a vertical sliding and the hinge motion of the transport domain. Computationally revealing such a global conformational change is unprecedented for elevator transporters.

## Metal binding facilitated IFC-to-OFC conformational switch

The MTD simulations showed a clear correlation between the direction of the metal movement and the elevator motion of the transport domain (Supplementary Table 2), but it is unclear whether the metal is the driving force for the elevator motion or whether it takes a free ride on a spontaneous conformational switch[53]. Previously, we have conducted a cysteine accessibility assay to detect the conformational states of BbZIP in the membrane fractions of *E.coli* cells[38]. In this work, we used this approach to investigate the role of metal substrates in determining the conformational states of the transporter. We chose two variants, L200C and A203C, because the single cysteine residue in each variant is accessible to the solvent only when the transporter is in the OFC (Fig. 5A). As shown in Fig. 5B, both BbZIP variants preferentially adopted the IFC in the presence of the metal substrates ($Zn^{2+}$ and $Cd^{2+}$), which was evidenced by less NEM labeling under the native conditions but more mPEG maleimide 5k (mPEG5k) labeling upon denaturation, when compared to the results obtained in the absence of metal. This result indicates that binding of metal substrates significantly increases the population of the IFC. In order to determine which metal binding site(s) are responsible for the metal induced conformational change, we disrupted the M1, M2, or M3 metal binding site in the L200C and A203C variants by replacing the metal chelating residues with alanine residues (Fig. 2). The resulting six variants, including ΔM1-L200C (H177A/E211A/L200C), ΔM1-A203C (H177A/E211A/A203C), ΔM2-L200C (N178A/D208A/E240A/L200C), ΔM2-A203C (N178A/D208A/E240A/A203C), ΔM3-L200C (H149A/H151A/L200C), and ΔM3-A203C (H149A/H151A/A203C), were applied to the cysteine accessibility assay (Fig. 5B). In sharp contrast to the variants with a deleted M2 site, disruption of the M1 or M3 site nearly eliminated the effects of metal binding on the accessibility of the introduced cysteine residues, indicating that the metal binding to M1 and M3 sites triggers the formation and maintenance of the IFC whereas metal binding to the M2 site does not. This result supports the notion

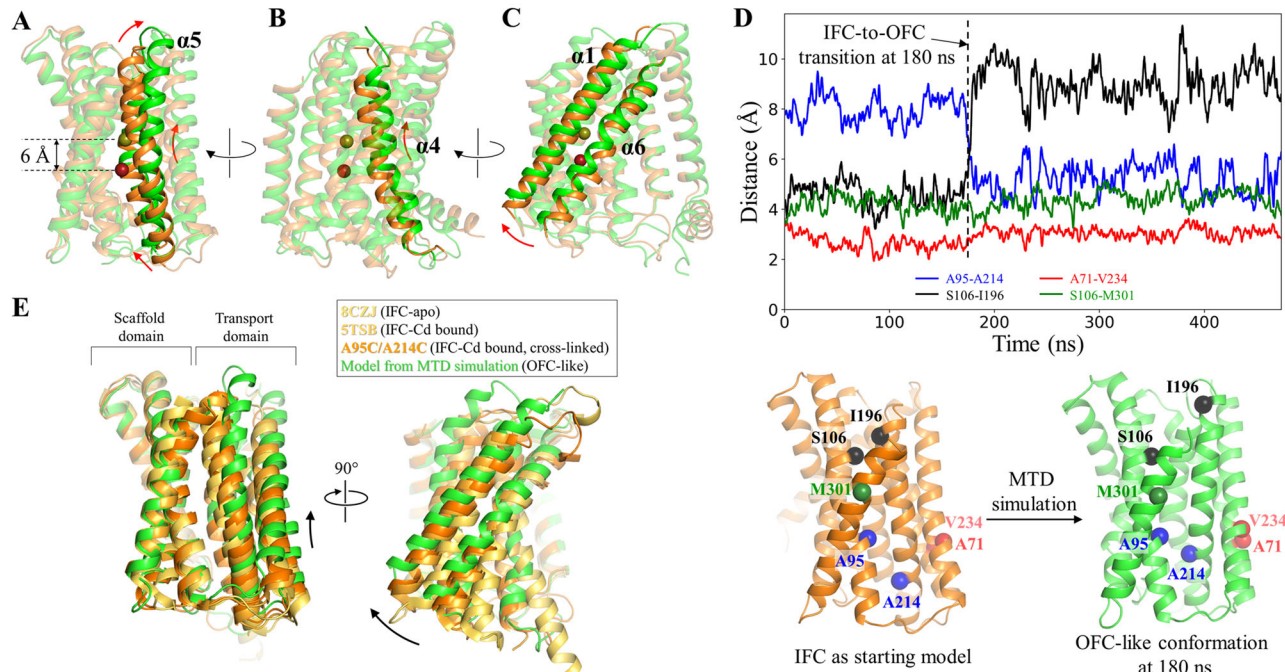

**Fig. 4 | The IFC-to-OFC transition uncovered in the MTD simulations.** Conformational changes of the highlighted structural elements – TM5 (**A**), TM4 (**B**), and TM1/TM6 (**C**). The dark red and yellow-green spheres represent Zn$^{2+}$ before and after the IFC-to-OFC transition, respectively. **D** Distance changes between the selected residue pairs throughout the simulation. The Cα atoms of the selected residues are shown as spheres and labeled in the initial IFC model and the OFC-like conformation generated in the simulation. The dashed line indicates the time when the IFC-to-OFC transition occurs at ~180 ns. **E** Combination of the vertical sliding (*left*) and the hinge motion (*right*) of the transport domain. The scaffold domains of the representative BbZIP structures are aligned to reveal the conformational changes of the transport domain. The results of the MTD simulations showing the similar IFC-to-OFC transition are shown in Fig. S10.

that the M1 site, but not the M2 site, is the authentic transport site[35] and that metal binding triggers the OFC-to-IFC transition of BbZIP through the charge-compensation mechanism[36,38], highlighting the importance of neutralization of transport site charges by the countercharged substrates in facilitating the elevator motion of the transport domain against the uncharged scaffold domain[54–57]. It is, however, unexpected that the M3 site, which is not a part of the transport site, is also crucially involved in the substrate induced conformational change. To further the understanding of the role of the M3 site, we incubated the H149A/H151A variant (where the M3 site is disrupted) in the Cd-bound state with excess Zn$^{2+}$ and determined the Cd/protein molar ratio after the treatment using ICP-MS. The result showed that the H149A/H151A variant barely retained any Cd$^{2+}$ (0.1 ± 0.03 Cd per protein, mean ± s.e.m., n = 2), whereas wild-type BbZIP with the same treatment still retained 1.3 ± 0.4 Cd per protein molecule (mean ± s.e.m., n = 2) (Fig. 5C), indicating that the variant is unable to retain Cd$^{2+}$ bound to the transport site in the presence of a competing substrate (Zn$^{2+}$). This is likely due to the increased metal release from the transport site, and thus the reduced metal binding to the transporter, upon disruption of the M3 site, which is consistent with the results of the functional study (Fig. 2F–H) and the MTD simulations (Fig. 3), and the proposed metal release pathways (Fig. 2K).

## Discussion

ZIPs are crucially involved in the homeostasis of a panel of essential trace elements and it is imperative to elucidate their working mechanisms because of the great potential applications in biomedicine, agriculture, and environmental protection. Although BbZIP, a prototype of the ZIP family, has been extensively studied in recent years, key questions regarding the transport mechanism, including the details of the proposed conformational changes and the translocation of the metal substrate through the transporter, remain to be answered. In this work, we addressed these issues by integrating the experimental and computational approaches to gain a thorough understanding of the transport mechanism, which not only reveals an unprecedented elevator transport mode, but also a highly regulated process of metal release from the transport site, providing a paradigm for studying other metal transporters.

### Metal release from the transport site into the cytoplasm

Three metal binding sites have been identified in BbZIP (Fig. 2), including the M1 and M2 sites at the transport site and the M3 site at the end of the metal release tunnel. For BbZIP to function as a metal transporter rather than a metal binding protein, it is critical to efficiently release the metal substrate from these high-affinity binding sites into the cytoplasm. Our experimental and computational studies revealed a metal release mechanism (Fig. 2K), which is likely conserved in the ZIP family, and also indicated that the metal release process is differentially regulated (Fig. 6).

At normal (or low) intracellular zinc levels, the metal at the M1 site can move to the unoccupied M3 site via Path 1 (Fig. 2K). Given that two carboxylic acid residues and one (or two) histidine residue are involved in metal chelation at the M3 site, it is conceivable that Zn$^{2+}$ will be held for a while before it is finally released into the cytoplasm. Indeed, the M3 site is one of the local minima along Path 1 (Fig. 3A and Supplementary Fig. S8A). Consistently, when the corresponding M3-forming residues in hZIP4 (except for D604, which is the equivalent to E276 in BbZIP and essential for the transport activity due to its multiple roles during metal release, Figs. 2H, I, K and 3A) were substituted with alanine residues, the zinc transport activities were increased (Fig. 2F, G). In this sense, the M3 site functions as a metal sink that negatively regulates the transport activity by transiently holding the metal substrate. For metal transporters, it is not rare that a cytoplasmic metal binding motif negatively regulates the transport activity. AtMTP1, a vacuolar Zn$^{2+}$/H$^+$ antiporter from *Arabidopsis thaliana* belonging to the cation diffusion facilitator superfamily, exhibited a

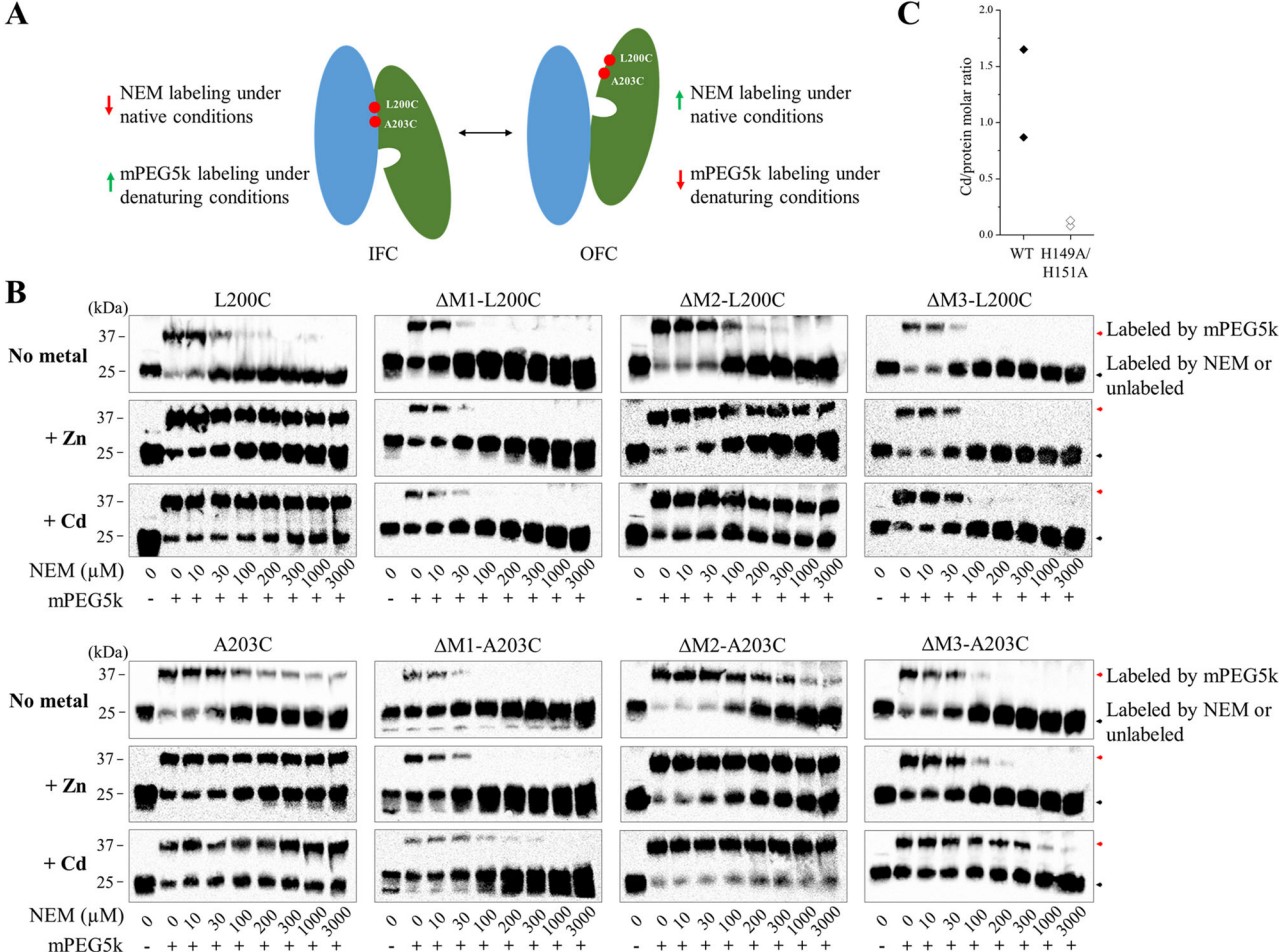

**Fig. 5 | Detection of metal binding induced conformational changes of BbZIP using cysteine accessibility assay. A** Illustration of cysteine labeling with NEM or mPEG5k when BbZIP is in the IFC or OFC. The scaffold domain and transport domain are colored in blue and green, respectively. **B** Cysteine accessibility assay of the variants in the presence or absence of metal substrates (50 μM). The membrane fractions of the *E.coli* cells expressing the variants were treated with the indicated amounts of NEM under native conditions, followed by the treatment of mPEG5k under denaturing conditions. At least three independent experiments were conducted with similar results. **C** Cd content determined by ICP-MS after the treatment of the Cd-bound proteins (wild-type BbZIP and the H149A/H151A variant) in a Cd-free solution (Fig. 2C) with excess zinc ions (250 μM). The Cd/protein molar ratios from two independent experiments are shown as rhomboids.

greatly increased activity when the cytoplasmic His-rich loop was deleted[58]. Similarly, substitution of the very C-terminal cytoplasmic "His-Cys-His" metal binding motif of the copper transporter CTR1 with alanine residues significantly increased both $K_M$ and turnover rate[59], reminiscent of the 5HA variant of hZIP4 (Fig. 2G). When the intracellular zinc is elevated to the level that the M3 site is saturated, the MTD simulations showed that $Zn^{2+}$ at the M1 site can be released into the cytoplasm only through Path 2 (Figs. 2K and 3C), which is however kinetically unfavorable when compared to Path 1. Therefore, the M3 site may act as an intracellular zinc sensor that auto-inhibits the activity of the transporter upon intracellular zinc overload[37].

Collectively, our data indicate that the transport activity can be differentially regulated by the M3 site through two mechanisms. It functions as a metal sink under normal conditions or a metal sensor for auto-inhibition upon zinc overload, acting overall as a negative regulator of the transport activity. This notion is further supported by the results of the cysteine accessibility assays (Fig. 5), which demonstrated that metal binding to the M3 site promotes the stability of the IFC, presumably by reducing the rate of metal release from the transport site. Since the structural elements enabling this dual function, including the conserved metal chelating residues (D144 and E276 in BbZIP) and the His-rich cluster in the IL2, are present in most ZIPs, it is likely that the unraveled mechanisms for

metal release and regulation are shared among ZIPs. Physiologically, a highly regulated transport, when combined with a slow transport rate[60], may help to control zinc flux more precisely as zinc is known to be a signaling molecule[61].

**An unusual elevator transport mode**
A hallmark of the elevator transport mode is the rigid-body sliding of the transport domain relative to the scaffold domain, which alternately exposes the transport site carried by the transport domain to the different sides of the membrane. For BbZIP, both the experimentally solved structures and the computational simulations showed that the transport domain can also undergo a hinge motion (Figs. 1E, 4 and Supplementary Fig. 10). As revealed in the MTD simulations, the combined vertical sliding and the hinge motion opens the otherwise blocked pathway in the IFC and exposes the transport site to the periplasm (Fig. 4 and Supplementary Fig. 10). This conformational change, as illustrated in Fig. 7 and Supplementary Movie 1, distinguishes BbZIP from other elevator transporters. A recent study found that the well-characterized elevator transporters share similar structural features (in particular in the transport domain), indicating that they are all homologous, whereas the dissimilarity of BbZIP in structure with these elevator transporters suggests that BbZIP likely utilizes a transport mode

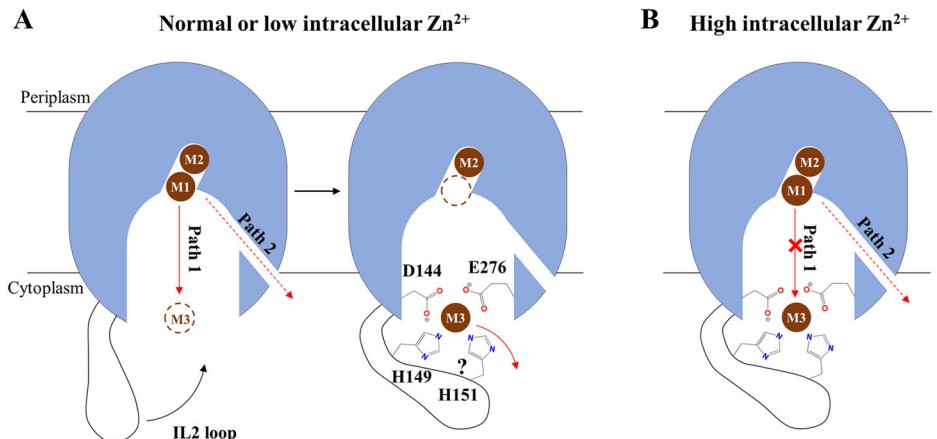

**Fig. 6 | Regulated metal release from the transport site into the cytoplasm. A** At normal or low zinc concentration, the metal released from the M1 site preferentially moves through Path 1 via the M3 site, where the histidine residue(s) in the IL2 participates in metal chelation, and is then slowly released into the cytoplasm. The question mark indicates that it is uncertain whether H151 is involved in binding $Zn^{2+}$ during the release process, as it is not suggested by MTD simulation. **B** At high intracellular zinc concentration, the M3 site is occupied by a zinc ion from the cytoplasm, which blocks Path 1. Metal can be released into the cytoplasm only through Path 2 at a slower rate. In either scenario, the IL2 negatively regulates the transport activity.

distinct from the canonical elevator mode[40]. Indeed, the substantial hinge motion seen in BbZIP is unprecedented in the previously studied elevator transporters[50,51], indicative of an unusual elevator mode. Given the long history of the ZIP family in evolution, the mechanism utilized by BbZIP, and potentially by other ZIPs, may represent an unprecedented ancient variant of the elevator transport mode.

In conclusion, our integrated structural, biochemical, and computational studies elucidated the mechanism for regulated substrate translocation through the transporter and revealed an unusual elevator-type conformational change, improving the understanding of the transport mechanism of the ZIP metal transporter family. In particular, our enhanced sampling simulations revealed how the transporter undergoes a metal-driven IFC-to-OFC switch. To the best of our knowledge, computationally discovering such a conformational change in simulations is unprecedented for elevator transporters. The application of a biasing potential on the substrate ($Zn^{2+}$) in the MTD simulations with optimized collective variables is key to achieving the global conformational switch, and we envision that this approach can be widely applied to other transporters, especially those whose major conformational changes are facilitated by substrate binding, as demonstrated in this study (Fig. 5).

## Methods

### Genes, plasmids, mutagenesis, and reagents

The DNA encoding BbZIP (National Center for Biotechnology Information reference code: WP_010926504) was synthesized with optimized codons for *E. coli* (Integrated DNA Technologies) and inserted into the pLW01 vector with a thrombin cleavage site inserted between the N-terminal His-tag and BbZIP. The DNA encoding human ZIP4 (GenBank access number: BC062625) from Mammalian Gene Collection were purchased from GE Healthcare, and inserted into a modified pEGFP-N1 vector (Clontech) in which the downstream EGFP gene was deleted and an HA tag was added at the C-terminus. Site-directed mutagenesis was conducted using the QuikChange® site-directed mutagenesis kit (Agilent) and verified by DNA sequencing. Oligos for mutagenesis are listed in Supplementary Data 1. 1-Oleoyl-rac-glycerol (monoolein), N-ethylmaleimide, 1,10-phenanthroline, Tris (2-carboxyethyl) phosphine (TCEP) were purchased from Sigma-Aldrich. mPEG-Maleimide MW 5k (#PLS-234) was purchased from Creative PEGworks.

### Protein preparation

Expression of the A95C/A214C variant of BbZIP was the same as reported[34,38]. In brief, the expression was conducted in the *E.coli* strain C41 (DE3) pLysS (Lucigen) in LBE-5052 autoinduction medium for 24 h at room temperature. After harvest, spheroplasts were prepared and lysed in the buffer containing 20 mM Hepes (pH 7.3), 300 mM NaCl, 0.25 mM $CdCl_2$, and the cOmplete protease inhibitors (Sigma-Aldrich). n-Dodecyl-β-D-maltoside (DDM, Anatrace) powder was added to solubilize the membrane fraction (final concentration 1.5%, w/v). The His-tagged protein was purified using HisPur Cobalt Resin (Thermo Fisher Scientific) in 20 mM Hepes (pH 7.3), 300 mM NaCl, 5% glycerol, 0.25 mM $CdCl_2$, and 0.1% DDM. The sample was then concentrated and loaded onto a Superdex Increase 200 column (GE Healthcare) equilibrated with the gel filtration buffer containing 10 mM Hepes, pH 7.3, 300 mM NaCl, 5% glycerol, 0.25 mM $CdCl_2$, and 0.05% DDM. 1 mM TCEP was included throughout the purification process except for the size exclusion chromatography. To reduce $Cd^{2+}$ level, the pooled peak fractions were briefly treated with 5 mM EDTA and immediately applied to a PD-10 desalting column (Cytiva) which has been equilibrated with the gel filtration buffer without $Cd^{2+}$. Careful control of the metal content to avoid metal oversaturation or metal loss from high-affinity binding sites was found to be critical to maintaining the metal-bound IL2 in its folded state. The protein was then mixed with $HgCl_2$ at a 1:10 molar ratio and allowed to incubate at room temperature for 30 min before setting up crystallization trays. To confirm a complete $Hg^{2+}$ cross-linking, the sample was analyzed by ICP-MS after desalting to remove free $Hg^{2+}$. The molar ratio of Hg:protein was found to be 0.95.

To conduct chemical crosslinking in the membrane fraction of the *E. coli* cells expressing wild-type BbZIP or its variants (A95C/A214C, A95C, and A214C), the spheroplast-derived membrane fraction of the cells from 1 liter culture was suspended in 6 ml solution containing 20 mM Hepes, pH 7.3, 300 mM NaCl, and 5% glycerol. 30 μL of the membrane fraction suspension was treated with the indicated concentrations of [Cu(II)(1,10-phenanthroline)]$_3$ at room temperature for 1 h. The membrane fraction was washed once to remove excess copper complex before it was dissolved in SDS-PAGE sample loading buffer and applied to a non-reducing SDS-PAGE. The samples were analyzed by Western blot using the custom mouse anti-BbZIP antibody reported previously[38] at 1:500 dilution for a stock at 0.03 mg/ml. and an HRP-conjugated anti-mouse immunoglobulin G antibody (Cell Signaling Technology, Catalog# 7076S, Lot 38) at 1:5000 dilution. The images of

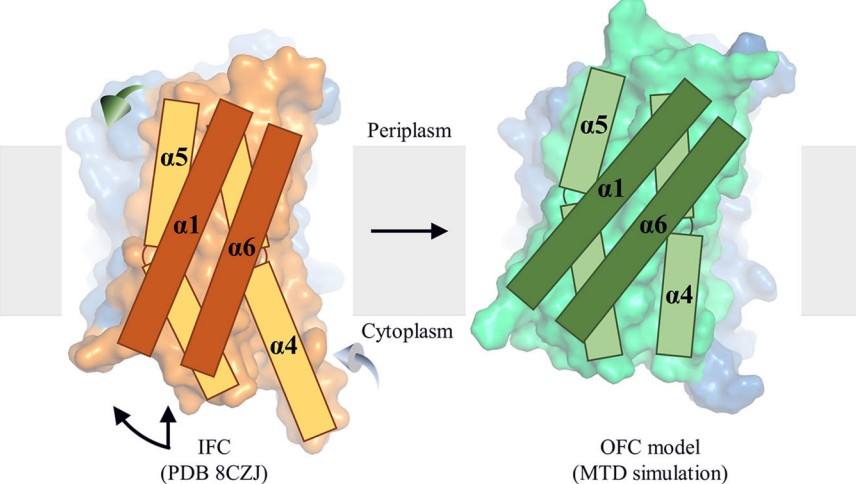

**Fig. 7 | Proposed conformational change during the IFC-to-OFC transition.** The scaffold domains (blue) are structurally aligned and the transport domains are shown in orange (IFC) or green (OFC). The transmembrane helices (TM1, TM4, TM5, and TM6) in the transport domain are shown as rectangles. To switch from the IFC to the OFC during a transport cycle, the transport domain undergoes a combined hinge motion (curved arrow) and a vertical sliding (straight arrow), along with the outward swing of the periplasmic portion of TM5 (green 3D arrow) and the inward swing of the cytoplasmic portion of TM4 (light blue 3D arrow).

the blots visualized by chemiluminescence (VWR) were taken using a Bio-Rad ChemiDoc Imaging System.

### Crystallization and structure determination

Purified A95C/A214C variant was concentrated to 15 mg/ml and then mixed with the molten monoolein with two coupled syringes at a ratio of 2:3 (protein/monoolein, v/v). All crystallization trials were set up using a Gryphon crystallization robot (Art Robbins Instruments). 50 nl of BbZIP-monoolein mixture covered with 800 nl of well solution was sandwiched with lipidic cubic phase sandwich set (Hampton Research). Stick-shaped crystals appeared after one week under the condition containing 30% PEG200, 100 mM Hepes, pH 7.0, 100 mM NaCl, 100 mM CaCl₂, at 21 °C and grew to full size in two weeks. Crystals were harvested with a MiTeGen micromesh and flash-frozen in liquid nitrogen. The X-ray diffraction data were collected at the General Medicine and Cancer Institutes Collaborative Access Team (GM/CA-CAT) (23-ID-B/D) at Advanced Photon Source (APS). The diffraction datasets were indexed, integrated, and scaled in HKL2000[62]. The apo state structure was solved in molecular replacement using the previously solved structure (PDB: 5TSB) as the search model in Phenix[63]. Iterative model building and refinement were conducted in COOT[64] and Phenix, respectively. All figures of protein structures were generated by PyMOL v1.3 (Schrödinger LLC).

### Mammalian cell culture and zinc transport assay

Human embryonic kidney cells (HEK293T, ATCC, Catalog# CRL-3216) were cultured in Dulbecco's modified eagle medium (DMEM, Thermo Fisher Scientific) supplemented with 10% (v/v) fetal bovine serum (FBS, Thermo Fisher Scientific) and Antibiotic-Antimycotic solution (Thermo Fisher Scientific) at 5% CO₂ and 37 °C. Cells were seeded on the polystyrene 24-well trays (Alkali Scientific) for 16 h in the basal medium and transfected with 0.8 µg DNA/well using lipofectamine 2000 (Thermo Fisher Scientific) in DMEM with 10% FBS.

The zinc transport activities of ZIP4 and the variants were tested using the cell-based transport assay. Twenty hours post transfection, cells were washed with the washing buffer (10 mM HEPES, 142 mM NaCl, 5 mM KCl, 10 mM glucose, pH 7.3) followed by incubation with Chelex-treated DMEM media (10% FBS). 5 µM Zn²⁺ (0.05 µCi/well) was added to cells. After incubation at 37 °C for 30 min, the plates were transferred on ice and the ice-cold washing buffer with 1 mM EDTA was added to stop metal uptake. The cells were washed twice and pelleted through centrifugation at 120 × g for 5 min before lysis with 0.5% Triton

X-100. A Packard Cobra Auto-Gamma counter was used to measure radioactivity. The transport activity was determined by subtracting the radioactivities of ⁶⁵Zn associated with the cells transfected with the empty vector from those associated with the cells transfected with metal transporters.

hZIP4-HA and the variants expressed on the plasma membrane were determined by cell surface bound anti-HA antibody as reported[65]. In brief, cells were washed twice with Dulbecco's phosphate-buffered saline (DPBS) on ice and then fixed with 4% Formaldehyde for 5 min. Cells were then washed three times in DPBS and incubated with 5 µg/ml anti-HA antibody (Invitrogen, Catalog# 26183) at 4 °C overnight. Cells were washed five times in DPBS to remove unbound antibodies, lysed in SDS-PAGE sample loading buffer, and eventually applied to SDS-PAGE and Western blot.

For Western blot, the samples mixed with the SDS sample loading buffer were heated at 96 °C for 10 min before loading on SDS-PAGE gel. The protein bands were transferred to PVDF membranes (Millipore). After being blocked with 5% nonfat dry milk, the membranes were incubated with a mouse anti-HA antibody at 1:5000 dilution (Invitrogen, Catalog# 26183, Clone 2-2.2.14, Lot XD345993) at 4 °C overnight. As loading control, β-actin levels were detected using a rabbit anti-β-actin antibody at 1:5000 dilution (Cell Signaling Technology, Catalog# 4970S, Clone 13E5, Lot 19). Primary antibodies were detected with an HRP-conjugated anti-mouse immunoglobulin-G at 1:5000 dilution (Cell Signaling Technology, Catalog# 7076S, Lot 38) for ZIP4 or an HRP-conjugated anti-rabbit immunoglobulin-G for β-actin at 1:5000 dilution (Cell Signaling Technology, Catalog# 7074S, Lot 30) by chemiluminescence (VWR). The images of the blots were taken using a Bio-Rad ChemiDoc Imaging System.

### Metal content measurement by ICP-MS

Wild-type BbZIP and the H149A/H151A variant were purified as described above and the N-terminal His₆-tag was removed by incubation with thrombin overnight at 4 °C before it was loaded to a Superdex Increase 200 column (GE Healthcare) equilibrated with a Cd-free solution containing 10 mM Hepes (pH 7.3), 300 mM NaCl, 5% glycerol, and 0.05% DDM. The peak fractions were pooled and the protein concentrations were measured using Bio-Rad protein assay. To study metal exchange with excess zinc, the Cd-bound wild-type BbZIP and the H149A/H151A variant prepared in the Cd-free solution were treated with 0.25 mM ZnCl₂ at room temperature for 30 min and then applied to a Superdex Increase 200 column (GE Healthcare) equilibrated with

10 mM Hepes (pH 7.3), 300 mM NaCl, 5% glycerol, 0.05% DDM, and 0.25 mM $ZnCl_2$. The peak fractions were pooled and the protein concentrations were measured using Bio-Rad protein assay. To prepare samples for ICP-MS analysis, 50 µl protein sample was mixed with 100 µl of 70% nitric acid (Fisher chemical, Cat# A509P212) in 15 ml metal-free tube (Labcon, Cat# 3134-345-001-9). The samples were heated in a 60 °C water bath for 1 h before being diluted to 3 ml using MilliQ water. Samples were analyzed using the Agilent 8900 Triple Quadrupole ICP-MS equipped with the Agilent SPS 4 Autosampler.

## Cysteine accessibility assay

The spheroplast-derived membrane fractions of the *E. coli* cells expressing the BbZIP variants with or without the treatment of 50 µM metal ($ZnCl_2$ or $CdCl_2$) were incubated with the indicated concentrations of NEM at 4 °C for 1 h, washed twice with a solution containing 100 mM Tris (pH 7.0), 60 mM NaCl, and 10 mM KCl (to remove excess NEM), and then dissolved in the denaturing solution containing 6 M urea, 0.5% SDS, and 0.5 mM DTT (to quench any residual NEM) by gentle shaking at room temperature for 15 min. The samples were then treated with mPEG5K at a final concentration of 5 mM to label the unmodified cysteine residue at room temperature for 1 h before they were mixed with 4xSDS-PAGE sample loading buffer containing 20% β-mercaptoethanol (β-ME) and subjected to SDS-PAGE. The custom mouse anti-BbZIP monoclonal antibody was used to detect BbZIP in Western blots and HRP-conjugated anti-mouse immunoglobulin G antibody was used as the secondary antibody at 1:5000 dilution (Cell Signaling Technology, catalog number 7076S). Images of the blots visualized by chemiluminescence (VWR) were taken using a Bio-Rad ChemiDoc imaging system.

## Computational simulations

Standard MD simulations the cryo-EM structure of BbZIP dimer was used as the initial model[37], and the cadmium ions were substituted with zinc ions. The equilibrated system was then compared to the Cd-bound structure, showing that the protein structure did not undergo significant conformational changes upon metal replacement (Supplementary Fig. 7B). Next, the optimized $C_4$ terms were incorporated into the system to enhance the interactions between the metal ions-ligands (carboxylic acid residues and histidine) and the metal ion-water complexes using the parameters developed by Merz group[47,48]. The recently developed $C_4$ terms for divalent metal ion-ligands can accurately reproduce the experimental free energy interactions between metal ions and the negatively charged residues (aspartate and glutamate), as well as the histidine residue[47,48]. The systems were generated using the CHARMM-GUI server[66], employing the TIP3P water model, with potassium and chloride used as counterions for system neutralization. To mimic the lipid bilayers of gram-negative bacteria, 200 lipid molecules, composed of a 3:1 ratio of POPE to POPG, were employed. Following this, a five-stage minimization process was carried out to optimize the simulation system. Afterwards, a 5 ns NVT ensemble simulation followed by a 5 ns NPT ensemble simulation was conducted to equilibrate the system. The equilibrated system was then subjected to for 2 µs standard MD simulations, and the entire trajectory was analyzed. The SHAKE[67] and Langevin dynamics algorithms were applied to constrain all covalent and hydrogen bonds and maintain the temperature at 303 K, respectively. Pressure control was achieved using isotropic pressure scaling and the Berendsen barostat[68]. All MD simulations were performed using the AMBER 22 package[69], applying both the ff19SB[70] and Lipid21[71] force fields. A time step of 0.002 ps and a nonbonded cutoff of 9 Å were applied in the simulations.

Steered molecular dynamics. The Zn-bound model derived from the cryo-EM structure of BbZIP dimer was generated as described above. Four rounds of minimization were carried out to reduce the system's energy, with 10,000 steps of the steepest descent used in each round. The system was then gradually heated to 303 K over 1 ns using a NVT ensemble. Following this, an NPT ensemble was employed for a 4 ns simulation. Then, 100 ns SMD simulations were conducted to pull the folded loop towards the cytosol, applying a force constant of 1000 kcal/(mol Å²). Bond lengths, including those involving hydrogen atoms, were constrained using the SHAKE algorithm[67]. Temperature regulation was achieved using the Langevin dynamics thermostat. A nonbonded cutoff distance of 10 Å was applied and Monte Carlo barostat[72] was employed to maintain the desired pressure by regulating the system volume. The simulations were performed with a time step of 0.002 ps and a nonbonded cutoff of 9 Å.

Metadynamics Simulations. Well-tempered (WT) volumetric-based metadynamics (MTD) simulation approach, using spherical coordinates of the metal ion (ρ, θ, and φ) with respect to the center of mass of the group of atoms comprising TM2, TM4, TM5, and TM7 as collective variables (CVs), was employed, following the methodology outlined in prior studies[46,73]. The WT-MTD method has been effectively used to uncover the detailed mechanism of action of the sodium-potassium-chloride cotransporter 1 (NKCC1)[46]. ρ represents the distance of the metal ion from the center of mass of the TMs mentioned above. θ is defined as the polar angle, and φ as the azimuthal angle. The movement of the metal ion away from its binding site was biased using parameters consistent with previous research (deposition time = 5 ps, hill height = 1.0 kJ/mol, $\sigma_{rho}$ = 0.5 Å, $\sigma_{theta}$ = π/16, $\sigma_{phi}$ = π/8, and bias factor = 10)[46]. Each WT-MTD simulation was repeated at least three times, with a duration of at least 500 ns. The sampling space of CV ρ was confined by restraining to 25 Å, corresponding to a distance where the metal ion can be found fully solvated outside the transporter.

In order to explain and better represent the metal ion movement along the transporter, an additional CV monitoring the coordination number (CN) of the zinc ion (group A) with the transporter heavy atoms (group B) has been employed. The formula used to define this CV is:

$$CN = \sum_{i \in A} \sum_{j \in B} \frac{1 - \left(\frac{r_{ij}}{r_0}\right)^n}{1 - \left(\frac{r_{ij}}{r_0}\right)^m} \tag{1}$$

Where a cutoff of $r_0$ = 4 Å has been used to define a contact, together with the powers being set to n = 8 and m = 16.

Free energy profiles were extracted after reweighting, which relied on two variables: the distance of the zinc ions from the reference center, labeled as CV1 (ρ), and the CN, defined above, as CV2.

All MTD simulation systems were equilibrated using the same method described in the Standard MD Simulations section. The density and temperature of each system were calculated during the last 1 ns of the equilibration phase to confirm that the system had equilibrated properly (Supplementary Figs. 12 and 13). The equilibrated systems were then used to perform the MTD production runs, with the lengths detailed in Table S2. The MTD analyses were performed on the entire trajectories. The PLUMED v.2.8 plugin[74,75] was used to conduct the MTD simulations. Detailed information about the simulation systems for each scenario and their convergence is presented in Supplementary Table 3 and Supplementary Fig. 14, respectively.

Using other biased sampling techniques, such as umbrella sampling, did not yield the desired results for the system under investigation in this study. Our umbrella sampling results indicated that the helices in contact with the metal ion either significantly or partially unfolded as the metal ion traversed the transporter, revealing very high energy barriers for the metal movement through the transporter.

Standard MD simulations after MTD simulations. To test whether the folded IL2 disrupts the closing of the cytoplasmic gate in the OFC-like conformation, the last snapshots of the relevant MTD simulations, in which $Zn^{2+}$ has been released into the periplasm, were taken and

used to perform standard MD simulations for 1 μs after unfolding of the IL2 in SMD as described above and energy minimization/equilibration.

## Reporting summary

Further information on research design is available in the Nature Portfolio Reporting Summary linked to this article.

## Data availability

The atomic coordinates and structure factors generated in this study have been deposited in the PDB with the accession code of 8J1M (Hg²⁺-crosslinked A95C/A214C variant of BbZIP). The atomic coordinates of other involved structures can be retrieved from PDB under the following access codes, including 5TSB (BbZIP in the Cd bound state), 5TSA (Zn²⁺-soaked Cd-bound BbZIP), 7Z6M (BbZIP in the Cd bound state), 7Z6N (BbZIP in the apo state), 8CZJ (BbZIP in the apo state), and 8GHT (cryo-EM structure of BbZIP dimer in the Cd bound state). No restrictions on the availability of any data or any material. Source data are provided with this paper.

## Code availability

AMBER is free for academic use and the source code can be found at https://ambermd.org. CPPTRAJ is included in the AmberTools software package, which is freely available at the AMBER website. The input, output, and parameter files required for the MD simulations in this study are available on Zenodo (https://doi.org/10.5281/zenodo.11103168). Additional files are available upon request from the corresponding authors. The analysis scripts and additional code are also available on Zenodo (https://doi.org/10.5281/zenodo.13910460).

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

## Acknowledgements

We thank the beamline scientists at GM/CA-CAT at APS for the assistance in data collections. We thank the following funding that supported

this work: NIH GM129004 and GM140931 (to J.H.), and GM130641 (to K.M.).

## Author contributions

J.H. and K.M. conceived and designed the project. Y.Z., T.Z., and D.S. conducted structural and biochemical experiments. M.J., and L.S. conducted computational simulations. Y.Z., M.J., T.Z., L.S., K.M., and J.H. analyzed the data and wrote the manuscript.

## Competing interests

The authors declare no competing interests.
