## [Transparent Peer Review file · Nature Communications]

Molecular insights into substrate translocation in an elevator-type metal transporter

Corresponding Author: Dr Jian Hu

Version 0:

Reviewer comments:

Reviewer #1

(Remarks to the Author)

The manuscript by Zhang et al. describes their work using crystallography, computation, and biochemistry to study the metal translocation process in a ZIP-family zinc transporter BbZIP. Using the structure of a Hg²⁺-cross-linked A95C/A214C mutant, they reveal an upward hinge motion, which could be an intermediate state toward an inward-facing state. In addition, they claim to identify a third metal binding site at the end of the transport pathway and capture the process of the elevator motion through simulation. Overall, their conclusions are well supported by the structure, simulation, and biochemistry data. The manuscript is also well-written and easy to follow. However, I have several concerns about the novelty of the discovery of this work, which may raise questions about the suitability of publishing the work in Nature Communications.

1) "An upward hinge motion of the transport domain reveals in the structure of a cross-linked transporter". Before this work, the same group and a group from Denmark proposed and elaborated the two-domain elevator model. In Fig 1E, the authors aligned five crystal structures of BbZIP including the current work. It's not clear what new data brought in from this Hg²⁺-cross-linked structure. Is this an additional structure to support the previous elevator model? Or the hinge motion does not exist from the comparison of previous structures alone? Looking at the trajectory as indicated by spheres in Fig. 1E, the hinge motion may be obtained with the previous data. In addition, these aligned structures have similar colors, making it difficult to see their similarity or differences. A detailed structural comparison of the Hg²⁺-cross-linked A95C-A214C mutant with previous structures is needed to understand the new features of this structure vs previous ones.

2) "Identification of the high-affinity metal binding sites". Compared to the previous BbZIP crystal structures, the cross-linked structure has the loop IL2 in an ordered conformation and a third metal binding site M3. The M3 site (E276, H151, His149, and D144) has been reported in a recent cryo-EM structure of the BbZIP dimer (ref 37). The authors didn't compare the two structures regarding the M3 site with an argument "While this work was under preparation, a cryo-EM structure of wild-type BbZIP revealed a similar M3 binding site." Nevertheless, the authors used the cryo-EM structure of the BbZIP dimer for all their computational work. The authors could have compared the crystal structure and the cryo-EM structure for the M3 site and report the results accordingly.

3) "A conserved metal release mechanism". The metal relay function of E276 has been previously proposed in ref 36 and elaborated in ref 37 by aligning structures involving metal binding sites, including the M3 site. The authors need to clarify the new findings from the Hg²⁺-cross-linked structure regarding a conserved metal release mechanism. In addition, the authors proposed two metal release pathways path 1 and path 2, assuming a folded IL2. However, there is no experimental data to show that IL2 can be folded without metal in the M3 site; in all M3-free structures, the IL2 is disordered. Therefore, the proposal and then disapproval of an unrealistic path 2 by computation, although increasing the amount of work, diminished the novelty of the work. To this reviewer, path 2 and the related simulation (scenario 2) should be removed from the manuscript.

Specific comments:

1) P. 3. "... upward rotation of the transport domain relative to the scaffold domain (Figure 1E)". The authors didn't give numbers of degrees and distance of translocation (for example relative to the M1 site) caused by the Hg²⁺-cross-linking. The authors are suggested to revise Fig. 1E to quantitatively present the results. These numbers also help to evaluate the uniqueness of this structure compared to previous crystal and cryo-EM structures of BbZIP.

2) "... the cross-linked A95C/A214C variant was briefly treated with EDTA". The authors used EDTA to remove Cd²⁺, but in the solved structure they found three Cd²⁺ sites M1, M2, and M3. Have they tried to use a longer treatment time or high EDTA concentration? It's also interesting to know why the M3 site exists in the cross-linked structure, but not in the metal-free structures of 8CZJ and 7Z6N and the metal-bound structure of 7Z6M. They should compare these crystal structures and explain.

3) P. 3. "... a continuous rigid-body hinge motion of the transport domain can be visualized (Figure 1E)". How different is the new structure compared with the rest structures in Figure 1E? Is it necessary to have this cross-linked structure to reach this conclusion? How far is this cross-linked structure (with the M1 site as a reference) relative to the proposed outward-facing conformation?

4) P. 3. "The occupancy analysis showed that the M1 site is more occupied (0.85) than the M2 site (0.42), indicating that the M1 site has a higher affinity than the M2 site, which is consistent with our previous report". In the authors' 2017 Sci Adv paper, they have shown that M1 has a higher affinity than the M2 site to Cd²⁺ and Zn²⁺. It's not clear what's new here, please explain.

The authors tried to use EDTA to remove Cd²⁺. If the occupancy for the M1 site is less than 1, there should be a state with the M1 site free of Cd²⁺. Are M1 free and M1 Cd²⁺ compatible in the same crystal structure?

5) P. 4. "While this work was under preparation, a cryo-EM structure of wild-type BbZIP revealed a similar M3 binding site". The Cryo-EM structure was published more than a year ago (Ref 37). The authors should compare their M3 site to the existing structure. What's the difference between the M3-site in crystal structure compared to that in the cryo-EM structure? Similarly, the role of H149 and H151A has been proposed and studied in the Cryo-EM paper. The crystal structure confirms the existence of the M3 site identified in the Cryo-EM structure. However, using a section titled "Identification of the high-affinity metal binding sites" is confusing. The authors should revise the section title and tune down their language on the work related to the M3 site.

6) P. 4. "To understand the role of E276 in BbZIP and the equivalent residues in other ZIPs, multiple metal bound structures of BbZIP (PDB: 5TSA and 7Z6M, and the Hg²⁺-cross-linked structure) were superimposed, revealing potential metal release pathways from the transport site to the cytoplasm here E276 plays a role in the metal relay". The metal relay role of E276 has been proposed in refs 36 and 37. The prior work should be cited.

7) P. 5. "The cryo-EM structure of wild-type BbZIP (PDB: 8GHT) was used as the initial structural model in the MTD simulations because it is the only experimentally solved structure of a BbZIP dimer". The Hg²⁺-cross-linked crystal structure is a monomer, while their MTD simulations used the dimer structure, presumably the monomeric BbZIP is not stable for simulation (Ref 49). It allowed the reviewer to question the importance of having the Hg²⁺-cross-linked structure in the computational work (Figs. 4-7).

8) P. 5. "... we conducted simulations in the following four scenarios". They simulated four scenarios to understand the metal release mechanisms, all starting with Zn²⁺ in the M1 site. Interestingly, in multiple simulations, they can find that Zn²⁺ releases to periplasm instead of cytoplasm. Did the authors try to simulate the uptake of Zn²⁺ starting from the periplasm, starting with the simulated or modeled outward-facing state? The proposed Scenario 2, illustrated in Fig. 2D, does not have any experimental structure data to support a folded metal-free IL2. Scenario 2 simulation should be removed.

9) Fig. 1E. It's difficult to match colors with respective PDBs in the alignment. It's suggested to use very different colors for each structure and provide detailed alignments for the new structure with each of the previous structures as Supplemental Figures.

10) Fig. 2. In panel A, at 4σ, it seems that the density for Hg²⁺ is between M1 and M2. What's the occupancy for Hg²⁺? Is it fully occupied? Did the authors have anomalous diffraction data to differentiate Hg²⁺ from Cd²⁺? It seems that the authors cannot remove Cd²⁺ in the M1 site using EDTA treatment. Is it possible the M1 site could be Hg²⁺? In panel B, it is hard to see the density of Cd²⁺ in the M3 site. What's the occupancy of Cd²⁺ in the M3 site? Have the authors analyzed the metal content for the EDTA-treated and Hg²⁺-cross-linked sample using ICP-MS?

11) Fig. 7. The left IFC and OFC models don't bring in new information compared to the right model. For simplicity and easy to follow, the authors should merge the two different diagrams into one to highlight the main findings of their work.

12) Fig. S2. How to tell that it is a Cl⁻, not something else? What are the B factor and occupancy for the modeled Cl⁻? Is that possible that the density could be for a partially occupied Hg²⁺ or partially occupied Cd²⁺?

13) Fig. S3. The states of IL2 and IL4 are experimentally verified in crystal and cryo-EM structures. It's not clear what's the purpose of this prediction using IUPred3, please explain more explicitly.

14) Fig. S4. The author should include a comparison of the Hg²⁺-cross-linked structure with the cryo-EM structure.

Reviewer #3

(Remarks to the Author)

The authors describe a combined structural biology and computational study into the ion release and substrate translocation

mechanism on the BpZIP metal transporter. Specifically, a novel structure of a cross-linked variant together with MD data revealed the ion release mechanism to the intracellular side. In addition, metadynamics simulations revealed an inward-facing to outward facing elevator conformational change, revealing a novel type of elevator transport mechanism. The manuscript is of high quality and well written. I have, however, a major concern that should be addressed before I can recommend it for publication.

The IFC to OFC conformational change appears to be based on a single observation in a series of metadynamics simulations. Although an intriguing observation, a single observation calls into question the statistical significance and reproducibility of the result. This transition should be investigated more thoroughly.

minor:

- I could not find the time step of the simulations in the methods section
- "exponentials" on p. 14 should read "powers"?

Reviewer #4

(Remarks to the Author)

Dear Authors

The report is interesting and new. I have, however, some comments I think could improve the manuscript:

- Manuscript should have line numbers.
- Page 2: "and an druggable node" should be "and a druggable node".
- Page 3: "has suggested that, when the transport": there is no comma here.
- Page 3: "some residue pairs that are far apart in the IFC approach in the OFC38, including A95 and A214 on the scaffold and transport domains, respectively": sentence seems to need complementation – what happens with the residues? They come closer, presumably, but it is important to make that explicit and clear in the text to readers.
- Page 3: despite the fact that the structure is referenced, this should be explained here – "in the cadmium (Cd) bound structure (PDB: 5TSB)".
- Figure 1C: please label the two gels with their respective labels.
- Results shown in Figure 2 about a third metal site are quite interesting and potentially new. I suggest authors compare it with what was published in the study that also described this new site (cited by authors at the beginning of section, but without much comparison to their own results). Moreover, it would be very important to increase sequence comparisons of conservation of that site in the wider ZIP family (authors seem to have only included a few in Fig 2E; ZIPs are widespread, and knowing whether this is common to several of them, in distantly related organisms, would add to the impact of the data.
- Page 5: please explain here a bit more the concept of metal sink in the context of transporters, and whether that is common/uncommon in other transporter folds.
- The molecular dynamics simulation results are quite compelling and new.
- Figure 6 (model): please separate in A and B.
- Page 9: authors claim that "Our experimental and computational studies revealed a metal release mechanism (Figure 2K), which is likely conserved in the ZIP family, and also indicated that the metal release process is differentially regulated (Figure 6)". I suggest authors provide a more thorough analyses of conservation of important residues and regions of ZIP members in several distantly related organisms, including fungi, plants, and animals. That would add to the claim, showing that sequence and structural necessary regions are present and this could be potentially common to ZIPs. I do not ask for experiments and simulations, but only for a sequence and structural comparative analyses to make that claim stronger.
- Again the claim "Given the long history of the ZIP family in evolution, the mechanism utilized by BbZIP, and potentially by other ZIPs, may represent an unprecedented ancient variant of the elevator transport mode" would be stronger if authors provided the data/analyses requested in the previous comment.
- Even in discussion section, authors provide little comparison between their data on M3 and the recently published paper. I request they do that, in order to make sure they agree or not with each other, and what is new on this particular site in this report.

Version 1:

Reviewer comments:

Reviewer #1

(Remarks to the Author)

I am pleased to read the revised manuscript by Zhang, et al. They have addressed most of my concerns. However, there are a few things that need to be clarified for me from the authors' response.

Additional questions to authors' response to my overall comments 1) and 3):

1) The authors responded that "From these structural comparisons, it can be concluded that the upward hinge motion cannot be obtained by using the previous structures." In the revised Fig. 1E, the authors used

Cd-bound state, i.e. blue, as a reference to define upward hinge motion relative to downward hinge motion. If we use apo structure, black, PDB 7Z6N, as a reference, may we reach an upward hinge motion? Please explain the rationale for using the Cd-bound state as a starting point. To this reviewer, logically, an apo-state IFC structure, i.e. after zinc release, could be a starting point toward OFC.

3) The reviewer appreciates the authors' explanation of performing MTD on scenario 2, which indeed helps to address the M3 sink function. What about scenario 3, where they wrote "the third scenario, the M1 and M2 sites were occupied by Zn²⁺ and the IL2 was in the folded state with a pre-formed but empty M3 site to test the accessibility of metal to an unoccupied M3 site upon a folded IL2" (lines 220-222). What's the rationale for performing MTD for a metal-free M3 site? Please explain in more detail.

It is interesting that Paths 1 and 2 could be used for zinc release based on the MTD simulations. Can authors verify experimentally that Path 2 is indeed feasible? What are the physiological conditions that Path 2 may be used for zinc uptake in BbZIP?

Additional questions to authors' response to my specific comments 1) and 12:

1) The authors responded that "the hinge motion from 5TSB to the new structure leads to a 0.2 Å shift of the M1 site". This is a very small change, considering a distance of 6 Å required from IFC to OFC based on their MTD calculations (Fig. 4A). In the simulation (Fig. 4D), the distance between S106 and I196 increased dramatically at ~180 ns. What's the distance between S106 and I196 in their Hg²⁺ cross-linked structure?

12) The authors argued that "a bound metal (Cd or Hg) was ruled out because it would be too close to Cd²⁺ at the M1 site (2.6 Å)" and "the distance between the density and Cd²⁺ at the M1 site matches the Cd-Cl bond length (~2.5 Å)". The authors also mentioned that the occupancy for the M1 Cd²⁺ is 0.85. Then, how can the authors rule out the possibility that the modeled Cl⁻ could be Cd²⁺ with an occupancy of 0.15? That means Cd²⁺ could be at the M1 site or the modeled Cl⁻ site. In Fig. S3, the side chains of H177 and Q207 may provide the needed coordination for a partially occupied Cd²⁺ or even Hg in the Cl⁻ site. The authors should revisit their analysis of the Cl⁻ site. Without validation of the Cl⁻ site next to the M1 site, they went too far to speculate the function of Cl⁻ in BbZIP (lines 120-125).

Reviewer #3

(Remarks to the Author)

The authors have satisfactorily addressed my concerns.

Reviewer #4

(Remarks to the Author)

Version 2:

Reviewer comments:

Reviewer #1

(Remarks to the Author)

The authors have addressed most of my concerns associated with their original manuscript. The assignment of a chloride ion to the density near the M1 Cd²⁺ remains arbitrary. In their 2017 Sci Adv paper (DOI: 10.1126/sciadv.1700344), they included 300 mM Cl⁻ for crystallization (100 mM NaCl, 100 mM CdCl₂, and 100 mM tris-HCl). However, in that case, they didn't observe a chloride ion in the M1 site. The authors should discuss the rationale of assigning Cl⁻, or remove the related text in the manuscript.

Response to reviewers' comments

Reviewer #1 (Remarks to the Author)

The manuscript by Zhang et al. describes their work using crystallography, computation, and biochemistry to study the metal translocation process in a ZIP-family zinc transporter BbZIP. Using the structure of a Hg₂⁺-cross-linked A95C/A214C mutant, they reveal an upward hinge motion, which could be an intermediate state toward an inward-facing state. In addition, they claim to identify a third metal binding site at the end of the transport pathway and capture the process of the elevator motion through simulation. Overall, their conclusions are well supported by the structure, simulation, and biochemistry data. The manuscript is also well-written and easy to follow. However, I have several concerns about the novelty of the discovery of this work, which may raise questions about the suitability of publishing the work in Nature Communications.

R: Thanks for the reviewer's comment that our conclusions are well supported by the data. We believe that this work, which combines the results of structural, computational, and biochemical studies, is novel for the following reasons.

First, the cross-linked structure reveals a novel inward-facing conformation (IFC) which represents an intermediate state during the elevator-like IFC-to-OFC transition. As shown in the updated Figure 1E, the cross-linked structure (red) shows an upward hinge motion (greater twist) when compared to the structure of the wild-type protein in the Cd-bound state (PDB 5TSB, blue), whereas the apo state structures (grey to black) show downward hinge motions (smaller twist). The new Figure S2 (next page) compares the new structure with 5TSB and the cryo-EM structure (PDB 8GHT), indicating a substantially altered orientation of the transport domain, which cannot be obtained by using the previous structures. By the time we conducted this research, a prominent

question was whether this twist (or the hinge motion) could be extrapolated to eventually allow the IFC-to-OFC switch. Our new structure provides a timely answer that the transport domain *can* undergo further hinge motion on the trajectory toward the OFC. This conclusion is supported by the disulfide bond formation between A95C and A214C (Figure 1B) and the MTD simulations (Figure 4E) where an IFC-to-OFC conformational change was captured during the simulations and the new structure was found to represent an intermediate state during the IFC-to-OFC transition.

Second, our MTD simulations provided important insights into two essential events in a transport cycle – the release of the metal substrate from the transport site into the cytoplasm and the IFC-to-OFC

conformational switch. We would like to emphasize that studying transition metal-protein interaction using MD simulations is technically challenging, and our ability to conduct this

research builds on our recent progress in the study of zinc-ligand interactions (refs 47, 48). For the event of metal release from the transport site, although the metal release pathway in BbZIP has been suggested in previous reports, it is only in this work that the detailed process of metal release from the transport site to the cytoplasm is investigated using both experimental and computational approaches (Figure 2 and Figure 3). For the IFC-to-OFC transition, computationally capturing such a global conformational change for a transporter is a rare achievement. In literature, people often rely on the experimentally solved IFC and OFC to simulate the process of global conformational change of a transporter. However, the lack of an experimentally solved OFC prevented us from doing so in this work. In fact, the IFC-to-OFC switch caught during the MTD simulations was unexpected because we did not apply biased forces on the transporter, but the resulting OFC agreed well with the observed hinge motion (Figure 1E and Figure 4E) and also indicated a vertical sliding, the hallmark of elevator transporters. We believe that our success in this work provides a paradigm for the computational study of other transporters, particularly transition metal transporters.

Compared to previous publications, the combined structural, computational and biochemical study in this work elucidated (1) the steps through which metal is released from the transport site to the cytoplasm; (2) the path through which the IFC-to-OFC switch is achieved; and (3) a novel mode of elevator-like conformational change consisting of a hinge motion and a vertical slide, which has not been reported for other elevator transporters.

Third, we proposed a new function for the M3 site. In ref 37, the M3 site was proposed to function as an auto-inhibitory site that is occupied and blocks the metal release pathway when intracellular zinc level is high. In this work, as summarized in Figure 6, we proposed that the M3 site is inhibitory but functions differently depending on the intracellular zinc level. At high zinc, the M3 site is occupied by intracellular zinc to block Path 1, forcing metal to pass through Path 2 that is deemed to be slow due to the requirement of greater conformational change of the transporter (Figure 3C); at normal or low zinc, metal released from the transport site is trapped at the M3 site (a free energy well shown in Figure 3A and metal coordination at the M3 site shown in Figure 3B) before it is eventually released into the cytoplasm. This new metal sink function during transport, which negatively regulates transport rate, has been supported by mutagenesis and transport assay (Figure 2).

Overall, this work, supported by a combination of structural, computational, and biochemical studies, fills several critical missing knowledge gaps and provides the most complete and comprehensive picture of metal translocation through BbZIP to date. The Hg-cross-linked structure is only part of the story, and the novelty of this work lies not only in this structure, but in the combined results of complementary approaches that converge in the conclusion.

1) “An upward hinge motion of the transport domain reveals in the structure of a cross-linked transporter”. Before this work, the same group and a group from Denmark proposed and elaborated the two-domain elevator model. In Fig 1E, the authors aligned five crystal structures of BbZIP including the current work. It’s not clear what new data brought in from this Hg²⁺-cross-linked structure. Is this an additional structure to support the previous elevator model? Or the hinge motion does not exist from the comparison of previous structures alone? Looking at the trajectory as indicated by spheres in Fig. 1E, the hinge

motion may be obtained with the previous data. In addition, these aligned structures have similar colors, making it difficult to see their similarity or differences. A detailed structural comparison of the Hg²⁺-cross-linked A95C-A214C mutant with previous structures is needed to understand the new features of this structure vs previous ones.

R: We apologize for the confusing color, and we have made the suggested changes in Figure 1E. As described in the response to the previous comment, the cross-linked structure (red) shows an upward hinge motion (larger twist) when compared to the structure of the wild type protein in the Cd-bound state (PDB 5TSB, blue), whereas the apo state structures (grey to black) show downward hinge motion (smaller twist). The new Figure S2 compares the new structure with 5TSB and the cryo-EM structure (PDB 8GHT), indicating a substantially altered orientation of the transport domain. From these structural comparisons, it can be concluded that the upward hinge motion cannot be obtained by using the previous structures.

2) “Identification of the high-affinity metal binding sites”. Compared to the previous BbZIP crystal structures, the cross-linked structure has the loop IL2 in an ordered conformation and a third metal binding site M3. The M3 site (E276, H151, His149, and D144) has been reported in a recent cryo-EM structure of the BbZIP dimer (ref 37). The authors didn’t compare the two structures regarding the M3 site with an argument “While this work was under preparation, a cryo-EM structure of wild-type BbZIP revealed a similar M3 binding site.” Nevertheless, the authors used the cryo-EM structure of the BbZIP dimer for all their computational work. The authors could have compared the crystal structure and the cryo-EM structure for the M3 site and report the results accordingly.

R: Thanks for the suggestion. We have changed the section title “Identification of the high-affinity metal binding sites” to “Characterization of the high-affinity metal binding sites” to avoid confusion. In Figure S2C (see above), the IL2 structures in the new structure and the cryo-EM structure are compared, showing the overall similar structural features of the IL2 and Cd coordination at the M3 site with minor differences. We have updated the manuscript as shown below. (Page 3, first paragraph)

“Recently, a cryo-EM structure of wild-type BbZIP showed a similar M3 binding site³⁷, and structural comparison of the two structures revealed some minor differences (Figure S2C), which is consistent with a flexible IL2 and dynamic metal binding at the M3 site.”

3) “A conserved metal release mechanism”. The metal relay function of E276 has been previously proposed in ref 36 and elaborated in ref 37 by aligning structures involving metal binding sites, including the M3 site. The authors need to clarify the new findings from the Hg²⁺-cross-linked structure regarding a conserved metal release mechanism. In addition, the authors proposed two metal release pathways path 1 and path 2, assuming a folded IL2. However, there is no experimental data to show that IL2 can be folded without metal in the M3 site; in all M3-free structures, the IL2 is disordered. Therefore, the proposal and then disapproval of an unrealistic path 2 by computation, although increasing the amount of work,

diminished the novelty of the work. To this reviewer, path 2 and the related simulation (scenario 2) should be removed from the manuscript.

R: The function of E276 in metal relay was first suggested in our early study (ref 34) that reads "... and the bound metals will be released to the cytoplasm through a chain of metal-chelating residues (H177, E276, H275, and D144), ...". In ref 36, E276 was suggested to play a role in metal release based on the altered growth pattern of the E276A variant in a zinc toxicity assay. The M3 site was not mentioned in that work. In ref 37, E276 was discussed based on extensive structural comparison, but no functional study was conducted on this residue. These early studies consistently proposed a role for E276 in metal release from the M1 site. In this work, we proposed two additional functions of E276, including (1) directing the metal from the transient metal binding site to the M3 site during transport; and (2) contributing to the high affinity of the M3 site (as a metal sink). We have clarified the role of E276 in the revised manuscript as below. (Page 5, first paragraph)

"First, it transiently holds the metal that leaves the M1 site with the flipped H177. This role has been suggested by our and other previous studies^{34,36,37}. Second, its structural flexibility, likely due to the conserved P279 and the high dynamics of the segment connecting TM7 and TM8 (Figure S5), allows it to direct the metal to the M3 site. Indeed, replacing P607 in hZIP4 (equivalent to P279 in BbZIP) with alanine reduced the transport activity by ~60% (Figure 2J). Third, it contributes to metal binding to the M3 site. As a high-affinity binding site (Figure 2C) that is within the transport pathway and negatively regulates the metal transport activity (Figures 2F & 2G), the M3 site can be described as a metal sink in addition to the proposed autoinhibitory site that acts only upon zinc overload³⁷."

Path 2 was ignored in early studies and only investigated in this work. When the paper reporting the cryo-EM structure claimed that the M3 site blocks the metal release pathway, one might note that the folded IL2 actually blocks only half of the tunnel connecting the transport site to the cytoplasm (as shown in Fig. 3 of ref 37). The immediate question is whether or not metal can be released through the other half of the tunnel (Path 2 in this work). When we solved the Hg-cross-linked structure with a high resolution (1.95 Å), the question became even more critical because Path 2 was found to be filled with water, indicative of a hydrophilic environment that would stabilize metal ions. Therefore, it is important to investigate whether metal in the transport site can be released through Path 2 when Path 1 is blocked by the occupied M3 site. To answer this question, we conducted MTD simulations in the second scenario where M1, M2 and M3 are occupied and the IL2 is folded. Thus, we cannot agree with the reviewer that the second scenario should be removed as it answered an important question. We emphasized this point in the revised manuscript. (Page 5, first paragraph)

"As Path 2 is filled with water (Figure 2D), can it be used as an alternative pathway for metal release to the cytoplasm when Path 1 is blocked by the occupied M3 site? We investigated this important question using computational approaches."

Also, we disagree that Path 2 is *unrealistic* because the metal was eventually released into the cytoplasm in the MTD simulations via this pathway, despite a higher energy barrier and

thus a slower rate. As summarized in Figure 6, Path 2 is unfavorable when Path 1 is available (left panel) but it can still be used as an alternative when Path 1 is blocked (right panel).

Specific comments:

1) P. 3. "... upward rotation of the transport domain relative to the scaffold domain (Figure 1E)". The authors didn't give numbers of degrees and distance of translocation (for example relative to the M1 site) caused by the Hg²⁺-cross-linking. The authors are suggested to revise Fig. 1E to quantitatively present the results. These numbers also help to evaluate the uniqueness of this structure compared to previous crystal and cryo-EM structures of BbZIP.

R: Thanks for the suggestion. Due to the limited space in Figure 1, we decided to present the quantitative analysis of the suggested structural comparison in Figures S2A & S2B (see above), where the rotation angles are measured to be in the range of 3.4-8.8 degrees at five positions. The cross-linked structure is compared only with the Cd-bound structures (crystal and cryo-EM structures) because the rotation angles are even larger when compared with the apo-state structures, as shown in the updated Figure 1E.

2) "... the cross-linked A95C/A214C variant was briefly treated with EDTA". The authors used EDTA to remove Cd²⁺, but in the solved structure they found three Cd²⁺ sites M1, M2, and M3. Have they tried to use a longer treatment time or high EDTA concentration? It's also interesting to know why the M3 site exists in the cross-linked structure, but not in the metal-free structures of 8CZJ and 7Z6N and the metal-bound structure of 7Z6M. They should compare these crystal structures and explain.

R: Thanks for this important question. We have tried to remove all metals using EDTA for structural studies but only found that prolonged treatment led to protein aggregation. In early Cd-bound structures, high concentrations of Cd (e.g., 200 mM CdCl₂ for 5TSB) were present in the crystallization solution and the solved structures did not show a folded and metal-bound IL2. We postulate that too much metal may compete for binding to the metal chelating residues in the IL2 when it is disordered so that metal can only bind to D144 and E276 but H149 and H151 in the IL2 cannot join the M3 site because it has already been saturated with bound metals. Therefore, it is crucial to control the metal content in solution to obtain a structure with a bound metal at the folded IL2. We have briefly discussed this point in the revised manuscript. (Page 11, second paragraph)

"Careful control of the metal content to avoid metal oversaturation or metal loss from high-affinity binding sites was found to be critical to maintaining the metal-bound IL2 in its folded state."

3) P. 3. "... a continuous rigid-body hinge motion of the transport domain can be visualized (Figure 1E)". How different is the new structure compared with the rest structures in Figure 1E? Is it necessary to have this cross-linked structure to reach this conclusion? How far is this cross-linked structure (with the M1 site as a reference) relative to the proposed outward-facing conformation?

R: We have updated Figure 1E and made Figure S2 to better show the differences between the new structure and other previously solved structures of BbZIP. As explained above, the new structure is the only one that shows an upward hinge motion (or greater twist) when compared to the earliest Cd-bound structure (PDB 5TSB), while the other structures only show downward hinge motion (or less twist). A critical question after several IFC structures were solved was whether the transport domain could undergo an upward motion to eventually allow an IFC-to-OFC switch. The new structure, together with the results of the MTD simulations (Figure 4) and the disulfide bond formation between C95 and C214 (Figure 1B), gave a confirmatory answer to this question, providing important evidence to define the process of the elevator-like conformational change.

As shown in Figure 4E, the cross-linked structure is on the trajectory from 5TSB to the OFC, but it is closer to 5TSB than to the OFC. Accordingly, the hinge motion from 5TSB to the new structure leads to a 0.2 Å shift of the M1 site and a 0.6 Å shift of the M2 site, consistent with the notion that the cross-linked structure is an IFC. A combination of a hinge motion and a vertical sliding is required to complete the IFC-to-OFC transition.

4) P. 3. “The occupancy analysis showed that the M1 site is more occupied (0.85) than the M2 site (0.42), indicating that the M1 site has a higher affinity than the M2 site, which is consistent with our previous report”. In the authors’ 2017 Sci Adv paper, they have shown that M1 has a higher affinity than the M2 site to Cd²⁺ and Zn²⁺. It’s not clear what’s new here, please explain. The authors tried to use EDTA to remove Cd²⁺. If the occupancy for the M1 site is less than 1, there should be a state with the M1 site free of Cd²⁺. Are M1 free and M1 Cd²⁺ compatible in the same crystal structure?

R: To address the reviewer’s concern, we have moved the occupancy information from the main text to the legend of Figure 2 and deleted the mentioned sentence from the main text to avoid misunderstanding.

An occupancy of 0.85 at the M1 site does not necessarily mean the presence of 15% BbZIP in the apo state after EDTA treatment. In lattice, protein may lose bound metal when the crystallization solution is metal free. Radiation damage during data collection can also reduce the nominal occupancy of bound ligands, including metals.

5) P. 4. “While this work was under preparation, a cryo-EM structure of wild-type BbZIP revealed a similar M3 binding site”. The Cryo-EM structure was published more than a year ago (Ref 37). The authors should compare their M3 site to the existing structure. What’s the difference between the M3-site in crystal structure compared to that in the cryo-EM structure? Similarly, the role of H149 and H151A has been proposed and studied in the Cryo-EM paper. The crystal structure confirms the existence of the M3 site identified in the Cryo-EM structure. However, using a section titled “Identification of the high-affinity metal binding sites” is confusing. The authors should revise the section title and tune down their language on the work related to the M3 site.

R: For clarification, we have uploaded the cross-linked structure to bioRxiv (April 21, 2023, <https://www.ncbi.nlm.nih.gov/pmc/articles/PMC10153219/>) before the cryo-EM structure was published (June 9, 2023), but we were unable to publish it yet. Nevertheless, we agree that the sentence may cause confusion and it has been changed to “*Recently*, a cryo-EM structure of wild-type BbZIP revealed a similar M3 binding site”.

As suggested by the reviewer, we have compared the M3 site of the two structures (Figure S2C), showing that the structures are similar with minor differences, which have been described in the figure legend. The IL2 structures are also slightly different, which is consistent with the notion that the IL2 is highly dynamic.

The section title has been changed to “Characterization of the high-affinity metal binding sites” to avoid confusion.

6) P. 4. “To understand the role of E276 in BbZIP and the equivalent residues in other ZIPs, multiple metal bound structures of BbZIP (PDB: 5TSA and 7Z6M, and the Hg²⁺-cross-linked structure) were superimposed, revealing potential metal release pathways from the transport site to the cytoplasm here E276 plays a role in the metal relay”. The metal relay role of E276 has been proposed in refs 36 and 37. The prior work should be cited.

R: Please check our response to comment #3.

7) P. 5. “The cryo-EM structure of wild-type BbZIP (PDB: 8GHT) was used as the initial structural model in the MTD simulations because it is the only experimentally solved structure of a BbZIP dimer”. The Hg²⁺-cross-linked crystal structure is a monomer, while their MTD simulations used the dimer structure, presumably the monomeric BbZIP is not stable for simulation (Ref 49). It allowed the reviewer to question the importance of having the Hg²⁺-cross-linked structure in the computational work (Figs. 4-7).

R: The cryo-EM structure is a more suitable starting model for MTD simulations than the cross-linked structure because (1) it shows a dimer; and (2) it can be nicely superimposed with the monomeric structure (Figure S7A). However, the water-filled Path 2 revealed in our high-resolution structure led to the critical question of whether or not metal can circumvent the blocked Path 1 (due to metal binding at the M3 site) to reach the cytoplasm via the hydrophilic Path 2. The answer to this question will determine the importance of metal binding at the M3 site. If Path 2 was an alternative pathway with a low energy barrier for metal release into the cytoplasm, then the claim that the M3 site is an auto-inhibitory site would not stand. In short, the Hg-cross-linked structure led to a logical and critical question that inspired us to study this important problem using computational approaches.

8) P. 5. “... we conducted simulations in the following four scenarios”. They simulated four scenarios to understand the metal release mechanisms, all starting with Zn²⁺ in the M1 site. Interestingly, in multiple simulations, they can find that Zn²⁺ releases to periplasm instead of cytoplasm. Did the authors try to simulate the uptake of Zn²⁺ starting from the periplasm, starting with the simulated or modeled outward-facing state? The proposed

Scenario 2, illustrated in Fig. 2D, does not have any experimental structure data to support a folded metal-free IL2. Scenario 2 simulation should be removed.

R: We did not simulate zinc uptake using the OFC model, but it is a good suggestion and will be the subject of our future study as it requires a large amount of effort and probably deserves a separate publication.

In the second scenario, all the three metal binding sites are occupied by zinc ions and the IL2 is folded to mimic the situation observed in the cross-linked and the cryo-EM structures. We are confused why the reviewer said “a folded metal-free IL2” for the second scenario. As explained earlier, the simulations in this scenario addressed the question of whether or not metal can reach the cytoplasm through Path 2 when Path 1 is blocked by the occupied M3 site, which is a very important question regarding the significance of the M3 site. We do not agree with the reviewer that this part should be removed.

9) Fig. 1E. It’s difficult to match colors with respective PDBs in the alignment. It’s suggested to use very different colors for each structure and provide detailed alignments for the new structure with each of the previous structures as Supplemental Figures.

R: We have updated Figure 1E by showing the new structure in red, 5TSB in blue, and other structures with smaller twist in grey to highlight the difference among these structures. New Figure S2 also includes direct comparisons with 5TSB and the cryo-EM structure.

10) Fig. 2. In panel A, at 4σ , it seems that the density for Hg^{2+} is between M1 and M2. What’s the occupancy for Hg^{2+} ? Is it fully occupied? Did the authors have anomalous diffraction data to differentiate Hg^{2+} from Cd^{2+} ? It seems that the authors cannot remove Cd^{2+} in the M1 site using EDTA treatment. Is it possible the M1 site could be Hg^{2+} ? In panel B, it is hard to see the density of Cd^{2+} in the M3 site. What’s the occupancy of Cd^{2+} in the M3 site? Have the authors analyzed the metal content for the EDTA-treated and Hg^{2+} -cross-linked sample using ICP-MS?

R: The metal at the M1 site is determined to be Cd^{2+} because: (1) Hg^{2+} strongly prefers a linear coordination as shown for the Hg^{2+} bound to C95 and C214 in the cross-linked structure, whereas Cd^{2+} prefers a tetrahedral or octahedral geometry that matches the observed distorted octahedral coordination sphere at the M1 site; (2) the bond distances between the metal and the coordinating atoms (O, N, and Cl⁻) match Cd^{2+} very well but not so if the metal was Hg^{2+} .

As mentioned in the main text, the occupancy of Cd^{2+} at the M3 site is 0.84 (in the section of “Characterization of high affinity metal binding sites”). The density of Cd^{2+} at the M3 site can be better inspected in Figure S4 (see below).

We did not measure the metal content of the EDTA-treated cross-linked sample, but we tested the sample after Hg cross-linking and desalting, showing that Hg was bound to the A95C/A214C variant at a molar ratio of 0.95:1 (Hg:protein). The brief EDTA treatment may remove some bound Hg and radiation damage may also be responsible for the low occupancy of Hg (0.36). The Hg content data are shown in the updated Methods and Materials. (Page 11, second paragraph)

“To confirm a complete Hg²⁺ cross-linking, the sample was analyzed by ICP-MS after desalting to remove free Hg²⁺. The molar ratio of Hg:protein was found to be 0.95.”

11) Fig. 7. The left IFC and OFC models don't bring in new information compared to the right model. For simplicity and easy to follow, the authors should merge the two different diagrams into one to highlight the main findings of their work.

R: The left panel in Figure 7 summarizes the findings of this work with details while the right cartoon plays an instructive role to indicate where the event in the left panel takes place in a complete transport cycle. We agree that this arrangement may lead to confusion and decided to remove the right panel to highlight the major findings in this work.

12) Fig. S2. How to tell that it is a Cl⁻, not something else? What are the B factor and occupancy for the modeled Cl⁻? Is that possible that the density could be for a partially occupied Hg²⁺ or partially occupied Cd²⁺?

R: The density was determined to be a Cl⁻ based on the following facts. First, as explained in the legend of Figure S3, a water molecule is excluded because of the otherwise very low B factor and strong positive density. Second, a bound metal (Cd or Hg) was ruled out because it would be too close to Cd²⁺ at the M1 site (2.6 Å). The lack of metal chelating residues around this density also does not support metal binding. Third, there is a total of 300 mM Cl⁻ in the crystallization solution (100 mM NaCl and 100 mM CaCl₂), while no other inorganic anions are present in the solution. Fourth, the distance between the density and Cd²⁺ at the M1 site matches the Cd-Cl bond length (~2.5 Å). Lastly, the B factor of Cl⁻ (43 with an occupancy of 0.84) is consistent with the B factors of the surrounding water molecules (35-50). The legend of Figure S3 has been updated accordingly.

13) Fig. S3. The states of IL2 and IL4 are experimentally verified in crystal and cryo-EM structures. It's not clear what's the purpose of this prediction using IUPred3, please explain more explicitly.

R: As reviewer suggested, we have removed the prediction result and replaced it with a B-factor putty image showing high flexibility of IL2 and IL4 in the cross-linked structure in Figure S5.

14) Fig. S4. The author should include a comparison of the Hg²⁺-cross-linked structure with the cryo-EM structure.

R: The two structures have been compared in Figure S2B.

Reviewer #3 (Remarks to the Author)

The authors describe a combined structural biology and computational study into the ion release and substrate translocation mechanism on the BpZIP metal transporter. Specifically, a novel structure of a cross-linked variant together with MD data revealed the ion release mechanism to the intracellular side. In addition, metadynamics simulations revealed an inward-facing to outward facing elevator conformational change, revealing a novel type of elevator transport mechanism. The manuscript is of high quality and well written. I have, however, a major concern that should be addressed before I can recommend it for publication.

The IFC to OFC conformational change appears to be based on a single observation in a series of metadynamics simulations. Although an intriguing observation, a single observation calls into question the statistical significance and reproducibility of the result. This transition should be investigated more thoroughly.

R: Thanks for the reviewer's positive comments on our work.

We are a little confused about the gist of the reviewer's comment. Is the reviewer asking if we saw the transition more than once in 16 runs (which we did), or is he/she suggesting that using an algorithm that drives this transition more frequently would be preferable? We address both of these interpretations below.

As shown in Table S2, a total of 16 MTD simulations were conducted in four different scenarios. The IFC-to-OFC switch was observed in four simulations in three scenarios (except for the one with an unfolded IL2). The detailed structural changes and comparisons are shown in Figure 4 and Figure S10. Despite small differences, these simulations consistently showed an elevator-like conformational change, leading to the backward movement of the metal into the periplasm. The relatively low probability for the IFC-to-OFC transition is consistent with the notion that this type of the global conformational change faces a high energy barrier for transporters. Even for a transporter that undergoes this transition on

shorter timescales than many other transporters, implying a lower energy barrier, a low frequency of the IFC-OFC transition was still observed in an MD simulation study (one observation per 14 μ s, see also Table S2 of ref 53).

In fact, we have already explored various techniques to optimize the parameters for this specific transporter. We now know that the use of the 12-6-4 LJ parameters is important. When we used the standard 12-6 LJ parameters for the same starting model, we observed the transition in only one replicate out of 24 metadynamics simulations (data to be published). Instead, using 12-6-4 LJ parameters, one could reproduce our results either by conducting metadynamics simulations for a few microseconds or by performing shorter simulations with multiple replicates, as we did in this study.

When both the IFC and OFC conformations are available, alternative methods can be used to investigate the transition more thoroughly and observe it more frequently. For example, in a recent study on the cation-chloride cotransporter NKCC1 researchers used classical molecular dynamics simulations coupled with deep-learning-guided enhanced sampling techniques. They applied the on-the-fly probability enhanced sampling (OPES) method to address complex dynamical phenomena and accelerate the sampling of the conformational space [1]. Given that our OFC model aligns nicely with the experimental data, we may consider using this approach to study the transition more thoroughly in our future work, but it would not affect our current results and conclusions. We appreciate the reviewer's suggestion.

minor:

-I could not find the time step of the simulations in the methods section

R: We have added the required information in the revised manuscript (page 14).

"A time step of 0.002 ps and a nonbonded cutoff of 9 Å were applied in the simulations."

-"exponentials" on p. 14 should read "powers"?

R: Changed.

References

Ruiz Munevar, M.J., Rizzi, V., Portioli, C., Vidossich, P., Cao, E., Parrinello, M., Cancedda, L., and De Vivo, M. (2024). Cation Chloride Cotransporter NKCC1 Operates through a Rocking-Bundle Mechanism. *J Am Chem Soc* 146, 552–566. 10.1021/jacs.3c10258.

Reviewer #4 (Remarks to the Author):

Dear Authors

The report is interesting and new. I have, however, some comments I think could improve the manuscript:

- Manuscript should have line numbers.

R: Line numbers are added.

- Page 2: "and an druggable node" should be "and a druggable node".

R: Thanks. It has been corrected.

- Page 3: "has suggested that, when the transport": there is no comma here.

R: Thanks. It has been corrected.

- Page 3: "some residue pairs that are far apart in the IFC approach in the OFC38, including A95 and A214 on the scaffold and transport domains, respectively": sentence seems to need complementation – what happens with the residues? They come closer, presumably, but it is important to make that explicit and clear in the text to readers.

R: Thanks for the suggestion. We provided more context as shown below. (Page 3, first paragraph)

"Our previous study has suggested that when the transport domain moves upward against the scaffold domain to switch from the IFC to the OFC, the distances between some residue pairs in the IFC become shorter in the OFC³⁸, such as A95 and A214 on the scaffold and transport domains, respectively (Figure 1A). This represents the dynamics of the transporter, which is essential for the alternating access mechanism, but it also poses a challenge for structural studies."

- Page 3: despite the fact that the structure is referenced, this should be explained here – "in the cadmium (Cd) bound structure (PDB: 5TSB)".

R: The mentioned sentence has been edited for clarity. (Page 3, first paragraph)

"Given that the C α distance between A95 and A214 (10.1 Å) in the IFC, as shown in the cadmium (Cd) bound structure (PDB: 5TSB, Figure 1A), is longer than that between two disulfide bonded cysteine residues (3.0-7.5 Å⁴¹), the formation of the C95-C214 disulfide bond is consistent with the proposed elevator transport mode^{36,38}"

- Figure 1C: please label the two gels with their respective labels.

R: Done.

- Results shown in Figure 2 about a third metal site are quite interesting and potentially new. I suggest authors compare it with what was published in the study that also described this new site (cited by authors at the beginning of section, but without much comparison to

their own results). Moreover, it would be very important to increase sequence comparisons of conservation of that site in the wider ZIP family (authors seem to have only included a few in Fig 2E; ZIPS are widespread, and knowing whether this is common to several of them, in distantly related organisms, would add to the impact of the data.

R: Thanks for the suggestion. We have added Figure S2 to compare the new structure with the previously solved structures. In Figure S2C, the M3 sites from the new structure and the cryo-EM structure are compared to show minor differences.

We have conducted a sequence alignment of 56 ZIPs from representative species (human, fly, worm, fungi, plant, and bacteria) to show the conservation of the involved residues. The results have been shown in Figure S6.

- Page 5: please explain here a bit more the concept of metal sink in the context of transporters, and whether that is common/uncommon in other transporter folds.

R: We added the following sentence to better describe the reason why we believe the M3 site functions as a metal sink.

"As a high-affinity metal binding site (Figure 2C) that is within the transport pathway and negatively regulates the metal transport activity (Figures 2F & 2G), the M3 site can be described as a metal sink in addition to the proposed autoinhibitory site that acts only upon zinc overload³⁷"

On Page 9 (last paragraph), we have provided two examples in which a cytoplasmic metal binding motif negatively regulates the metal transport activity, including AtMTP1 from the cation diffusion facilitator superfamily and human CTR1 (a copper importer). Potentially, the his-rich loop in AtMTP1 and the HCH motif in CTR1 play a role as a metal sink.

- The molecular dynamics simulation results are quite compelling and new.

R: Thanks for the positive comments.

- Figure 6 (model): please separate in A and B.

R: Thanks for the suggestion. Figure 6 has been divided into A and B.

- Page 9: authors claim that "Our experimental and computational studies revealed a metal release mechanism (Figure 2K), which is likely conserved in the ZIP family, and also indicated that the metal release process is differentially regulated (Figure 6)". I suggest authors provide a more thorough analyses of conservation of important residues and regions of ZIP members in several distantly related organisms, including fungi, plants, and animals. That would add to the claim, showing that sequence and structural necessary regions are present and this could

be potentially common to ZIPs. I do not ask for experiments and simulations, but only for a sequence and structural comparative analyses to make that claim stronger.

R: Thanks for the suggestion. As mentioned above, we have conducted a sequence analysis of 56 ZIPs from representative species (human, fly, worm, fungi, plant, and bacteria) and the results (Figure S6) show that the positions of D144 and E276 are mostly occupied by carboxylate residues and that proline is frequently present at the positions equivalent to P279. The presence of a histidine-rich segment in the IL2 is a known feature for the ZIP family (refs 2, 3). Therefore, the proposed metal release mechanism is likely a shared feature in the ZIP family.

- Again the claim “Given the long history of the ZIP family in evolution, the mechanism utilized by BbZIP, and potentially by other ZIPs, may represent an unprecedented ancient variant of the elevator transport mode” would be stronger if authors provided the data/analyses requested in the previous comment.

R: Agree and thanks for the suggestion.

- Even in discussion section, authors provide little comparison between their data on M3 and the recently published paper. I request they do that, in order to make sure they agree or not with each other, and what is new on this particular site in this report.

R: Thanks for the suggestion. In the revised manuscript, the M3 site in this work is compared to the previously reported M3 site in Figure S2C, showing similar structural features with some minor differences in metal coordination and side chain orientation (described in the legend of Figure S2C).

Response to Review

Reviewer #1 (Remarks to the Author):

I am pleased to read the revised manuscript by Zhang, et al. They have addressed most of my concerns. However, there are a few things that need to be clarified for me from the authors' response.

Additional questions to authors' response to my overall comments 1) and 3):

1) The authors responded that “From these structural comparisons, it can be concluded that the upward hinge motion cannot be obtained by using the previous structures.” In the revised Fig. 1E, the authors used Cd-bound state, i.e. blue, as a reference to define upward hinge motion relative to downward hinge motion. If we use apo structure, black, PDB 7Z6N, as a reference, may we reach an upward hinge motion? Please explain the rationale for using the Cd-bound state as a starting point. To this reviewer, logically, an apo-state IFC structure, i.e. after zinc release, could be a starting point toward OFC.

R: We understand the concern of the reviewer. Since the cross-linked structure reported in this work falls in the trajectory between the OFC and the IFC, it addresses the question of how the transport domain in the IFC moves toward the OFC. Because of the advance made in this work, we are now able to map the conformational change beyond the boundary defined by the previously solved structures. The use of the Cd-bound structure (PDB 5TSB) as reference in Figure 1E serves to highlight that the cross-linked structure reveals a new conformational state characterized by an upward hinge motion when compared to all previously solved structures. Thus, we believe that the claim of "an upward hinge motion" is appropriate. As such a motion distinguishes BbZIP (and probably other ZIPs as well as) from other elevator transporters, we think that it deserves special emphasis.

To avoid misunderstanding, the following sentence in the legend of Figure 1E has been updated.

*“The arrows indicate the opposite directions of the hinge motion from the Cd-bound state **only to highlight the new conformation of the cross-linked structure when compared to the previously solved structures.**”*

3) The reviewer appreciates the authors' explanation of performing MTD on scenario 2, which indeed helps to address the M3 sink function. What about scenario 3, where they wrote “the third scenario, the M1 and M2 sites were occupied by Zn²⁺ and the IL2 was in the folded state with a pre-formed but empty M3 site to test the accessibility of metal to an unoccupied M3 site upon a folded IL2” (lines 220-222). What's the rationale for performing MTD for a metal-free M3 site? Please explain in more detail.

R: As we and others have shown that the M3 site is one of the high-affinity binding sites, one question is whether the M3 site is pre-formed as an unoccupied pocket (note that the lack of a folded IL2 in early solved structures does not necessarily exclude this possibility) or is formed only when metal comes. Our MTD simulations showed that metal from the M1 site couldn't enter the pre-formed M3 site. This result indicates that the formation of the M3 site, and probably as well as the folding of IL2, is induced by metal binding, consistent with a mechanism of “induced fit” rather than “conformational selection” when it comes to protein-ligand interaction. We believe that this information is helpful to understand the formation of the metal-bound M3 site.

It is interesting that Paths 1 and 2 could be used for zinc release based on the MTD simulations. Can authors verify experimentally that Path 2 is indeed feasible? What are the physiological conditions that Path 2 may be used for zinc uptake in BbZIP?

R: Thanks for the suggestion. As shown in Figure 6B, we propose that Path 2 is used when Path 1 is blocked by Zn^{2+} at high intracellular zinc concentration. Based on our kinetic study (Figure 2G), there is no sign that the zinc transport activity of ZIP4 is reduced when the concentration of the extracellular zinc was increased to as high as 50 μM . If higher concentrations of zinc and prolonged incubation were applied to HEK293 cells, the zinc-induced ZIP4 degradation (*JBC*, 2007, 282:6992) would complicate data interpretation. So the current cell-based transport assay is not suitable to test Path 2, but we envision that a proteoliposome-based transport assay may help to address this issue. Another challenge is that zinc ion is not coordinated with any metal chelating residue when it leaves E276 and enters the cytoplasm through Path 2 (Figure 2K). As a result, we cannot identify the Path 2-specific residues for site-directed mutagenesis to experimentally test the contribution of Path 2 in zinc transport.

Additional questions to authors' response to my specific comments 1) and 12:
1) The authors responded that “the hinge motion from 5TSB to the new structure leads to a 0.2 Å shift of the M1 site”. This is a very small change, considering a distance of 6 Å required from IFC to OFC based on their MTD calculations (Fig. 4A). In the simulation (Fig. 4D), the distance between S106 and I196 increased dramatically at ~180 ns. What's the distance between S106 and I196 in their Hg²⁺ cross-linked structure?

R: The distance between the C α atoms of S106 and I196 in the cross-linked structure is 7.2 Å. In 5TSB and 8CJZ (the IFC with and without Cd), the distances are 6.4 Å and 5.1 Å, respectively. In the OFC model derived from the MTD simulation, the distance increases to about 9 Å. Thus, the cross-linked structure is on the trajectory from the IFC to the OFC, but is closer to the IFC, which is consistent with the small shift of the M1 site.

12) The authors argued that “a bound metal (Cd or Hg) was ruled out because it would be too close to Cd²⁺ at the M1 site (2.6 Å)” and “the distance between the density and Cd²⁺ at the M1 site matches the Cd-Cl bond length (~2.5 Å)”. The authors also mentioned that the occupancy for the M1 Cd²⁺ is 0.85. Then, how can the authors rule out the possibility that the modeled Cl⁻ could be Cd²⁺ with an occupancy of 0.15? That means Cd²⁺ could be at the M1 site or the modeled Cl⁻ site. In Fig. S3, the side chains of H177 and Q207 may provide the needed coordination for a partially occupied Cd²⁺ or even Hg in the Cl⁻ site. The authors should revisit their analysis of the Cl⁻ site. Without validation of the Cl⁻ site next to the M1 site, they went too far to speculate the function of Cl⁻ in BbZIP (lines 120-125).

R: We found that the distance between Cl⁻ and the N ϵ atom of H177 is 3.29 Å and the distance between Cl⁻ and the O ϵ atom of Q207 is 3.47 Å, neither of which is short enough to support favorable binding if Cl⁻ were replaced by Cd²⁺. Since the M1 site is a high-affinity metal binding site, it is hard to imagine how such a very weak binding site could compete with a 15:85 ratio. Therefore, we do not favor the idea that Cl⁻ is actually a partially occupied Cd²⁺. In the revised manuscript, we have made changes to more accurately describe the situation associated with Cl⁻. (page 3, last paragraph)

“... a chloride ion (Cl⁻) was assigned to a density that is 2.6 Å away from Cd²⁺ at the M1 site (Figure S3). As there is no evidence to support metal co-transport with Cl⁻ and there are no residues that

interact with Cl^- directly, it is unlikely that Cl^- is a substrate of BbZIP. Rather, we speculate that the anion may assist ...”

Reviewer #3 (Remarks to the Author):

The authors have satisfactorily addressed my concerns.

R: Thanks!

Reviewer #1 (Remarks to the Author):

The authors have addressed most of my concerns associated with their original manuscript. The assignment of a chloride ion to the density near the M1 Cd²⁺ remains arbitrary. In their 2017 Sci Adv paper (DOI: 10.1126/sciadv.1700344), they included 300 mM Cl⁻ for crystallization (100 mM NaCl, 100 mM CdCl₂, and 100 mM tris-HCl). However, in that case, they didn't observe a chloride ion in the M1 site. The authors should discuss the rationale of assigning Cl⁻, or remove the related text in the manuscript.

R: We deleted the sentences about the potential function of the putative chloride ion and made further changes as shown below. (pages 3-4)

*"The coordination spheres of the bound Cd²⁺ are not significantly affected by the rigid-body hinge motion of the transport domain, except that a density 2.6 Å away from the Cd²⁺ at the M1 site was found (**Supplementary Figure 3**). This density, which excludes the sulfur atom of M99 from the coordination sphere of the M1 metal (**Figure 2A**), was assigned to be a chloride ion (Cl⁻) based on the crystallization condition and the surrounding contacts, but we cannot rule out the possibility that it may be a different Cd²⁺ ligand."*

The following sentence has been deleted in the legend of Figure 2.

"Note that a chloride ion (green sphere) displaces the sulfur atom of M99 in 5TSB from the coordination sphere at the M1 site."